# SARS-CoV-2 entry into human airway organoids is serine protease-mediated and facilitated by the multibasic cleavage site

**Anna Z Mykytyn[1†], Tim I Breugem[1†], Samra Riesebosch[1], Debby Schipper[1], Petra B van den Doel[1], Robbert J Rottier[2], Mart M Lamers[1‡], Bart L Haagmans[1‡*]**

[1]Viroscience Department, Erasmus University Medical Center, Rotterdam, Netherlands; [2]Department of Pediatric Surgery, Erasmus University Medical Center - Sophia Children's Hospital, Rotterdam, Netherlands

**Abstract** Coronavirus entry is mediated by the spike protein that binds the receptor and mediates fusion after cleavage by host proteases. The proteases that mediate entry differ between cell lines, and it is currently unclear which proteases are relevant in vivo. A remarkable feature of the severe acute respiratory syndrome coronavirus 2 (SARS-CoV-2) spike is the presence of a multibasic cleavage site (MBCS), which is absent in the SARS-CoV spike. Here, we report that the SARS-CoV-2 spike MBCS increases infectivity on human airway organoids (hAOs). Compared with SARS-CoV, SARS-CoV-2 entered faster into Calu-3 cells and, more frequently, formed syncytia in hAOs. Moreover, the MBCS increased entry speed and plasma membrane serine protease usage relative to cathepsin-mediated endosomal entry. Blocking serine proteases, but not cathepsins, effectively inhibited SARS-CoV-2 entry and replication in hAOs. Our findings demonstrate that SARS-CoV-2 enters relevant airway cells using serine proteases, and suggest that the MBCS is an adaptation to this viral entry strategy.

**\*For correspondence:**
b.haagmans@erasmusmc.nl

[†]These authors contributed equally to this work
[‡]These authors also contributed equally to this work

**Competing interests:** The authors declare that no competing interests exist.

## Introduction

The ongoing coronavirus disease (COVID-19) pandemic is caused by the severe acute respiratory syndrome coronavirus 2 (SARS-CoV-2), which emerged in central China late 2019 (*Zhu et al., 2020*). Within months, this virus spread globally, and as of October 15, 2020, over 38 million cases have been reported, including over 1 million deaths. Halting SARS-CoV-2 spread has shown to be highly complex, putting great strain on health systems globally. SARS-CoV-2 is the third zoonotic coronavirus to emerge from animal reservoirs within the past two decades, after SARS-CoV and Middle East respiratory syndrome coronavirus (MERS-CoV), in 2002 and 2012, respectively (*Drosten et al., 2003*; *Kuiken et al., 2003*; *Peiris et al., 2003b*; *Zaki et al., 2012*). In contrast to SARS-CoV-2, SARS-CoV and MERS-CoV have not attained sustained human-to-human transmission. These coronaviruses belong to the Betacoronavirus genus (family Coronaviridae, subfamily Orthocoronavirinae), which is thought to ultimately originate from bats, but can spread to humans via intermediate hosts (*Hu et al., 2015*; *Lau et al., 2010*; *Wang et al., 2014*).

Currently, it is largely unknown what factors determine coronavirus transmission to and between humans, but one important determinant may be the coronavirus spike (S) protein, which is the main glycoprotein incorporated into the viral envelope. Enveloped viruses, including coronaviruses, deposit their genomes into host cells by coalescing their membranes with the cell. This function is executed by S protein trimers, which fuse viral and cellular membranes after binding to the entry receptor (*Hulswit et al., 2016*). In addition, coronaviruses can spread from cell to cell when

coronavirus S proteins traffic to the plasma membrane of infected cells and fuse with neighboring cells, generating multinucleated giant cells (syncytia). Coronavirus S proteins are synthesized in infected cells in a stable and fusion-incompetent form and are activated through cleavage by host proteases. Proteolysis controls the timely release of the S protein's stored energy required to fuse membranes, which allows virions to be stable in the environment yet fusogenic after contacting entry receptors on host cell membranes.

Cleavage is essential for coronavirus infectivity and can occur in the secretory pathway of infected cells or during viral entry into target cells (*Hulswit et al., 2016*; *Millet and Whittaker, 2015*). Several groups of host proteases, including type II transmembrane serine proteases (hereafter referred to as serine proteases), proprotein convertases, and cathepsins, can cleave the S protein. Specific sites in the S protein regulate protease usage and therefore play an important role in determining cell tropism. Similarly, tropism can be determined by the availability of proteases that can activate the S protein (*Belouzard et al., 2012*; *Menachery et al., 2020*; *Millet and Whittaker, 2015*; *Yang et al., 2014*; *Yang et al., 2015*). The S protein consists of two domains, the receptor binding (S1) domain and the fusion (S2) domain. These domains are separated by the S1/S2 cleavage site, which in some coronaviruses, such as SARS-CoV-2, forms an exposed loop that harbors multiple arginine residues and is therefore referred to as a multibasic cleavage site (MBCS) (*Walls et al., 2020*; *Wrapp et al., 2020*). Cleavage of this site can occur in secretory systems of infected cells by proprotein convertases, including furin. S1/S2 cleavage does not directly trigger fusion but may facilitate or regulate further cleavage (*Park et al., 2016*). A second proteolysis step takes place at a more C-terminal site within the S2 domain, notably the S2′ site. S2′ cleavage is thought to occur after the virus has been released from producing cells and is bound to host cell receptors on receiving cells. The S2′ site is processed by serine proteases on the plasma membrane or by cathepsins in the endosome. Whereas S2′ cleavage appears to be crucial for coronavirus infectivity, not all coronaviruses contain a multibasic S1/S2 site and little is known of its function (*Hulswit et al., 2016*). Until recently, all viruses within the clade of SARS-related viruses, including SARS-CoV, were found to lack a multibasic S1/S2 cleavage site. However, SARS-CoV-2 contains a proline-arginine-arginine-alanine (PRRA) insertion into the S protein, precisely N-terminally from a conserved arginine, creating a multibasic RRAR cleavage motif. Exchanging the SARS-CoV-2 S MBCS for the SARS-CoV monobasic site was recently shown to decrease fusogenicity on a monkey kidney cell line (VeroE6) and infectivity in a human lung adenocarcinoma cell line (Calu-3) (*Hoffmann et al., 2020a*; *Shang et al., 2020*). However, cancer cells often poorly represent untransformed cells and thus the question remains whether the MBCS would affect infectivity on relevant airway cells. Another study showed that entry of SARS-CoV-2 pseudoparticles (PP) into Calu-3 cells could be blocked using a clinically approved serine protease inhibitor (camostat mesylate) (*Hoffmann et al., 2020b*). In contrast, in most other cell lines (including the lung cell line A549), recent CRISPR-Cas9 screens have suggested that SARS-CoV-2 entry in cell lines appears to be mediated by endosomal cathepsins (*Daniloski et al., 2020*; *Wei et al., 2020a*). These discrepancies between human lung cell lines underscore the importance of using relevant cells, such as human airway organoids (hAOs), to study SARS-CoV-2 entry. Here, we investigated if the SARS-CoV-2 MBCS affects entry into relevant human airway cells (1), if the MBCS can alter protease usage during entry (2), and which entry pathway is taken by SARS-CoV-2 in hAOs (3).

## Results

### Entry into lung adenocarcinoma cells and hAOs is facilitated by the SARS-CoV-2 S MBCS

Recently, *Hoffmann et al., 2020a* showed that the SARS-CoV-2 multibasic cleavage motif increases entry into Calu-3 cells by exchanging this motif and several N-terminally flanking amino acids with the monobasic S1/S2 site found in SARS-CoV or in a related bat virus RaTG13 (*Hoffmann et al., 2020a*). Building on these observations, we generated several SARS-CoV-2 S protein mutants and used these to generate vesicular stomatitis virus (VSV)-based PP stocks expressing a green fluorescent protein (GFP). Instead of exchanging cleavage sites, we mutated the minimal RXXR multibasic cleavage motif by deleting the PRRA insertion (Del-PRRA), changing the last arginine to an alanine (R685A) or to a histidine (R685H) in order to preserve the positive charge at this site (*Figure 1A*).

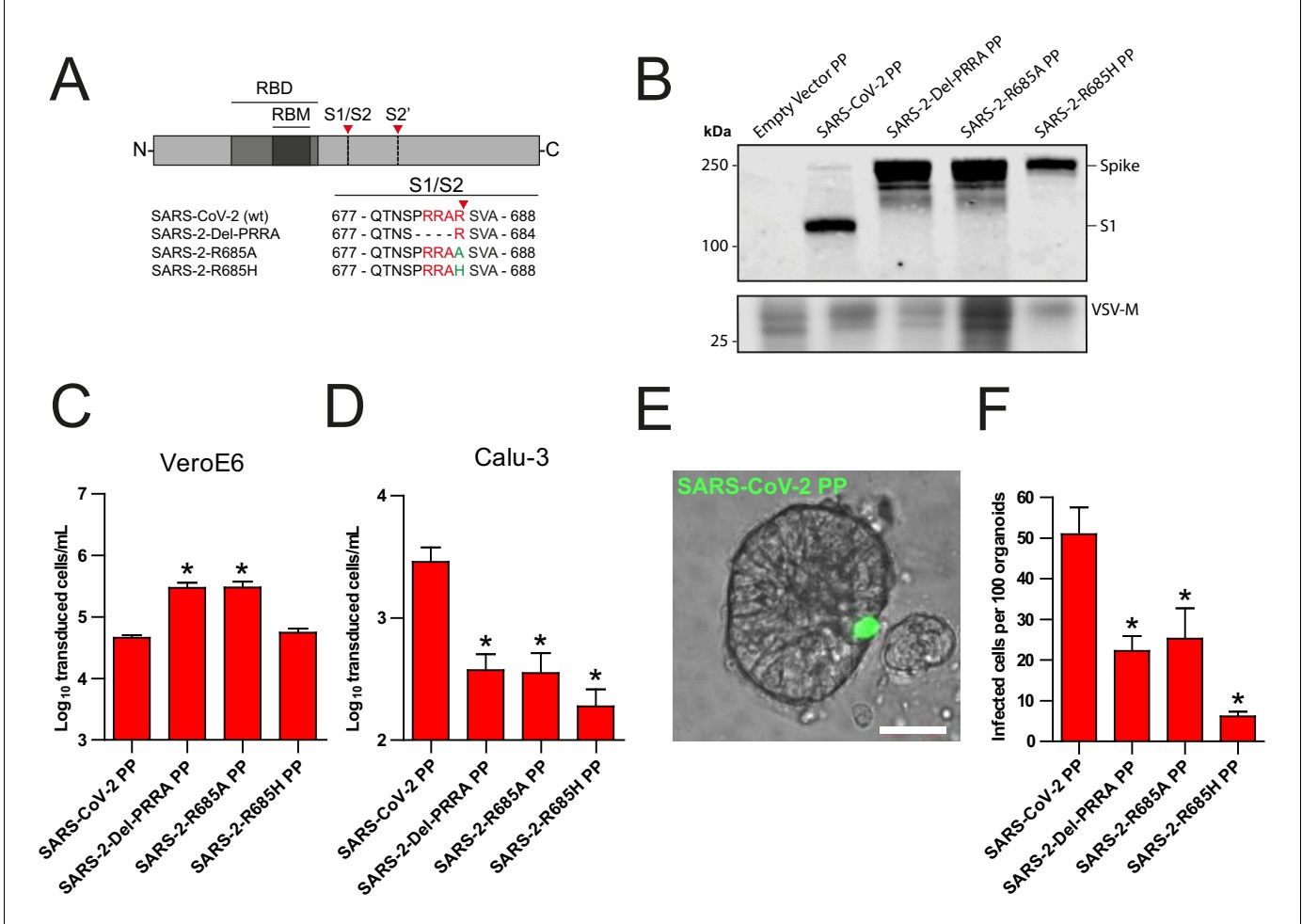

**Figure 1.** The SARS-CoV-2 S multibasic cleavage site mediates entry into human airway organoids. (**A**) Schematic overview of SARS-CoV-2 S protein mutants. MBCS residues are indicated in red; amino acid substitutions are indicated in green. Red arrows indicate cleavage sites. The SARS-CoV-2 S MBCS was mutated to either remove the PRRA motif (SARS-2-Del-PRRA) or substitute the R685 site (SARS-2-R685A and R685H). (**B**) Comparison of S cleavage of SARS-CoV-2 PPs and the MBCS mutants. Western blots were performed against S1 with VSV-M silver stains as a production control. (**C and D**) PP infectivity of SARS-CoV-2 S and MBCS mutants on VeroE6 (**C**) and Calu-3 (**D**) cells. (**E**) Differentiated hAO cultures were infected with concentrated SARS-CoV-2 PPs containing a GFP reporter, indicated in green. Scale bar indicates 20 μm. (**F**) SARS-CoV-2 PP and MBCS mutant infectivity on bronchiolar hAO cultures. One-way ANOVA was performed for statistical analysis comparing all groups with SARS-CoV-2 PPs. Experiments were performed in triplicate (**B and D, F**). Representative experiments from at least two independent experiments are shown (**C and D**). Combined data from three independent experiments is shown (**F**). Error bars indicate SEM. *$p < 0.05$. GFP, green fluorescent protein; hAO, human airway organoid; MBCS, multibasic cleavage site; PP, pseudoparticles; RBD, receptor binding domain; RBM, receptor binding motif.

The online version of this article includes the following figure supplement(s) for figure 1:

**Figure supplement 1.** hAO cultures grown at 2D air–liquid interface are well-differentiated and express ACE2 and TMPRSS2.

Immunoblotting revealed that wild-type and mutant PPs were produced at similar levels (*Figure 1B*). S1/S2 cleavage was observed for the wild-type SARS-CoV-2 PPs and abrogated by the PRRA deletion and the R685A and R685H substitutions, which is in agreement with studies showing that the SARS-CoV-2 S is cleaved by proprotein convertases, possibly furin (*Hoffmann et al., 2020a*; *Shang et al., 2020*). Next, we assessed the infectivity of these viruses and found that the SARS-2-Del-PRRA and SARS-2-R685A mutants were 5- to 10-fold more infectious on VeroE6 cells (*Figure 1C*). In contrast, on the lung adenocarcinoma cell line Calu-3, the SARS-2-Del-PRRA, SARS-2-R685A, and SARS-2-R685H PPs were approximately 5- to 10-fold less infectious compared with the wild-type PPs (*Figure 1D*). These data show that the PRRA deletion and single point mutations could functionally destroy the MBCS and suggest that this site enhances airway cell entry. Next, we assessed the effect of the MBCS in a relevant cell culture system. A 3D airway organoid model that

allows efficient SARS-CoV-2 replication where infectious virus titers increase over time has not yet been established, but we previously reported that 2D airway organoid-derived air–liquid interface differentiated cultures allow efficient SARS-CoV-2 replication (*Lamers et al., 2020*). In this model, hAOs (*Sachs et al., 2019*) are dissociated, seeded onto collagen-coated Transwell inserts, and differentiated at air–liquid interface in Pneumacult ALI medium (Stemcell). After differentiation, cultures contain ciliated cells, club cells, and goblet cells (*Figure 1—figure supplement 1*). Moreover, they express the SARS-CoV-2 entry receptor angiotensin-converting enzyme 2 (ACE2) and transmembrane protease serine 2 (TMPRSS2), a serine protease previously shown to mediate SARS-CoV-2 entry when overexpressed (*Figure 1—figure supplement 1D,E*; *Hoffmann et al., 2020b*). To set up a 3D model, we dissociated these 2D air–liquid differentiated cultures into small clumps, infected these in suspension, and then re-plated the clumps into basement membrane extract (BME), in which they formed spheroids. SARS-CoV-2 PPs successfully infected these hAOs, as observed by fluorescent microscopy (*Figure 1E*). SARS-CoV-2 PPs were approximately two times more infectious on these cells compared with the SARS-2-Del-PRRA and SARS-2-R685A mutants, and eight times more infectious than the SARS-2-R685H mutant (*Figure 1F*), demonstrating that the SARS-CoV-2 MBCS facilitates entry into human airway cells.

## SARS-CoV-2 enters Calu-3 cells faster than SARS-CoV and entry speed is increased by the MBCS

As SARS-CoV lacks the MBCS, we compared its infectivity to SARS-CoV-2 and found that both PPs readily infected Calu-3 cells, indicating that the SARS-CoV S has adaptations other than the MBCS to facilitate airway cell infection (*Figure 2A*). Likewise, inserting the PRRA motif into SARS-CoV S and thereby generating an MBCS did not increase PP infectivity on Calu-3 cells (*Figure 2—figure supplement 1*). To investigate this further, we compared the entry route taken by these viruses in Calu-3 cells. For this purpose, we used inhibitors of two major coronavirus entry pathways (*Hulswit et al., 2016*). Serine proteases are known to mediate early coronavirus entry on the plasma membrane or in the early endosome, whereas cathepsins facilitate entry in late, acidified endosomes. Concentration ranges of either a serine protease inhibitor (camostat mesylate; hereafter referred to as camostat) or a cathepsin inhibitor (E64D) were used to assess the entry route into Calu-3 cells. Entry of SARS-CoV-2 PPs was not inhibited by E64D, but could be inhibited by camostat, indicating that SARS-CoV-2 exclusively uses serine proteases for entry into these cells (*Figure 2B,C*). For SARS-CoV PPs, entry was inhibited slightly by E64D (~10%), but camostat had a far stronger effect (~90%), indicating that SARS-CoV mainly uses serine proteases to enter Calu-3 cells but that a small fraction of virions enter via cathepsins (*Figure 2B,C*). Previously, Calu-3 cells have been suggested to have low levels of cathepsin activity (*Park et al., 2016*). The observation that some SARS-CoV PPs use cathepsins suggests that this virus less efficiently uses the surface serine proteases encountered early during entry, resulting in particles accumulating in the endosome, where they are cleaved by cathepsins. To test this, we measured the serine protease-mediated entry rate of SARS-CoV-2 and SARS-CoV by blocking entry on Calu-3 cells at different time points postinfection using camostat. Cells were pretreated with E64D to prevent any cathepsin-mediated entry. Using both PPs and authentic virus (*Figure 2D,E*), we observed that SARS-CoV-2 entered faster than SARS-CoV via serine proteases. Next, we assessed whether the presence of an MBCS could increase the serine protease-mediated entry rate into Calu-3 cells. For this purpose, we used SARS-CoV S PPs containing the PRRA insertion (SARS-PRRA) (*Figure 2F*). Immunoblotting revealed that, in contrast to SARS-CoV PPs, SARS-PRRA PPs were partially cleaved (*Figure 2G*). Whereas wild-type SARS-CoV PPs used cathepsins, SARS-PRRA PPs did not (*Figure 2H,I*). The serine protease-mediated entry rate of SARS-PRRA PPs on Calu-3 cells was higher compared with SARS-CoV PPs (*Figure 2J*), and it was lower for SARS-2-Del-PRRA PPs compared with SARS-CoV-2 PPs (*Figure 2K*). These findings show that the SARS-CoV-2 MBCS facilitates serine protease-mediated entry on Calu-3 cells.

## Cell–cell fusion is facilitated by the SARS-CoV-2 MBCS and SARS-CoV-2 is more fusogenic than SARS-CoV on hAOs

Next, we used a GFP complementation cell–cell fusion assay (*Figure 3—figure supplement 1A,B*) to determine whether entry rate was associated with fusogenicity. In this assay, S and GFP-11 co-

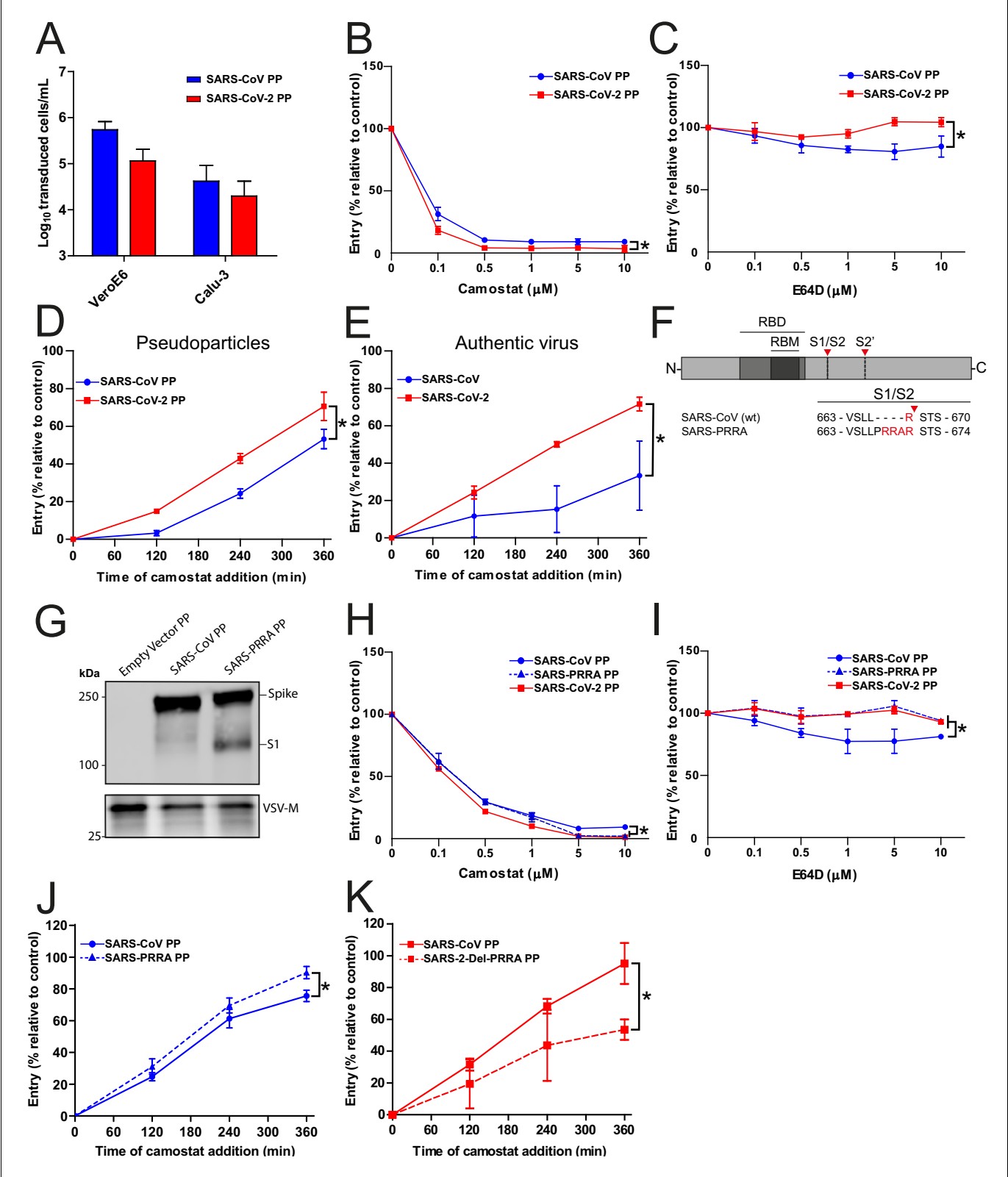

**Figure 2.** SARS-CoV-2 enters faster on Calu-3 cells than SARS-CoV and entry speed is increased by the multibasic cleavage site. (**A**) SARS-CoV PP and SARS-CoV-2 PP infectivity on VeroE6 and Calu-3 cells. (**B and C**) SARS-CoV PP and SARS-CoV-2 PP entry route on Calu-3 cells. Cells were pretreated with a concentration range of camostat (**B**) or E64D (**C**) to inhibit serine proteases and cathepsins, respectively. T-test was performed for statistical analysis at the highest concentration. *$p<0.05$. (**D and E**) SARS-CoV PP, SARS-CoV-2 PP (**D**) and authentic virus (**E**) entry speed on Calu-3 cells. T-test

*Figure 2 continued on next page*

*Figure 2 continued*

was performed for statistical analysis at the latest time point. *p<0.05. (F) Schematic overview of SARS-CoV S protein mutants. MBCS residues are indicated in red. The SARS-CoV-2 PRRA motif was inserted into SARS-CoV PPs (SARS-PRRA). (G) Comparison of S1 cleavage of SARS-CoV PP and the MBCS mutant. VSV-M silver stains are shown as a production control. (H and I) SARS-CoV PP, SARS-PRRA PP, and SARS-CoV-2 PP entry route on Calu-3 cells. Cells were pretreated with a concentration range of camostat (H) or E64D (I) to inhibit plasma membrane and endosomal entry, respectively. One-way ANOVA was performed for statistical analysis comparing all groups with SARS-CoV PPs at the highest concentration. *p<0.05. (J and K) Entry speed on Calu-3 cells of SARS-CoV PPs compared with SARS-PRRA PPs (J) and SARS-CoV-2 PPs compared with SARS-2-Del-PRRA PPs (K). T-test was performed for statistical analysis at the latest time point. *p<0.05. Experiments were performed in triplicate (A–E, H–K). Representative experiments from at least two independent experiments are shown. Error bars indicate SD. ANOVA, analysis of variance; MBCS, multibasic cleavage site; PP, pseudoparticles.

The online version of this article includes the following figure supplement(s) for figure 2:

**Figure supplement 1.** SARS-CoV PP infectivity into Calu-3 cells is not altered by the insertion of the multibasic cleavage site.

transfected HEK-293T cells fuse with GFP1-10 expressing Calu-3 cells, resulting in GFP complementation and fluorescence. In HEK-293T cells, MBCS containing S proteins were more cleaved than S proteins without this site (*Figure 3A*). We observed that SARS-CoV-2 S was more fusogenic than SARS-CoV S on Calu-3 cells (*Figure 3B,C*), VeroE6 cells, and VeroE6-TMPRSS2 cells (*Figure 3—figure supplement 1C–D and E–F*). The expression of TMPRSS2 increased fusion about twofold for both S proteins (*Figure 3—figure supplement 1G*). The insertion of the MBCS into SARS-CoV S increased fusion, whereas mutations in the SARS-CoV-2 S MBCS decreased fusion (*Figure 3B,C*). To investigate differences in fusogenicity in a relevant cell system, we infected 2D differentiated hAO air–liquid interface cultures with SARS-CoV-2 and SARS-CoV and assessed the formation of syncytial cells at 72 hr postinfection using confocal microscopy. Cells were termed syncytial cells when at least two nuclei were present within a single viral antigen-positive cell that lacked demarcating tight junctions. SARS-CoV-2 frequently induced syncytia, whereas SARS-CoV-infected cells rarely contained multiple nuclei (*Figure 3D,E* for quantification).

## The SARS-CoV-2 MBCS increases serine protease usage and decreases cathepsin usage

The findings above indicate that SARS-CoV-2 S is more fusogenic and mediates faster entry through serine proteases compared with SARS-CoV, indicating that the MBCS alters protease usage. To investigate this, cells that contain both serine and cathepsin protease-mediated entry should be used. Therefore, we focused on VeroE6 cells, which have an active cathepsin-mediated cell entry pathway, as on these cells both SARS-CoV-2 PP and SARS-CoV PP entry was inhibited by E64D, and not by camostat (*Figure 4A,B*). To generate a cell line in which both entry pathways are active, we stably expressed TMPRSS2 in VeroE6 cells. In these cells, SARS-CoV-2 PP entry was inhibited ~95% by camostat, whereas SARS-CoV PPs were only inhibited ~35% (*Figure 4C,D*). In accordance, E64D did not block SARS-CoV-2 PP entry, while it decreased SARS-CoV PP entry ~30%. These findings indicate that despite a functional serine protease-mediated entry pathway, a significant part of SARS-CoV PPs still retained cathepsin-mediated entry, whereas SARS-CoV-2 PPs only used serine proteases for entry. This phenotype was found to be linked to the MBCS as SARS-CoV-2 PPs containing mutations in this site entered less through serine proteases and more through cathepsins (*Figure 4E,F*). In accordance, the introduction of the MBCS into SARS-CoV PPs increased serine proteases usage, while decreasing cathepsin usage (*Figure 4G,H*).

## SARS-CoV-2 entry and replication are dependent on serine proteases in hAOs

Altogether, our findings show that SARS-CoV-2 preferentially uses serine proteases for entry, when present, and that the MBCS increases fusogenicity and infection of human airway cells. Hence, serine protease inhibition could be an attractive therapeutic option. Therefore, we assessed whether camostat could block SARS-CoV-2 entry and replication using hAOs. In these differentiated organoids the apical side of the cells was facing outwards (*Figure 5A,B*), facilitating virus excretion into the culture medium. These cells were infected with SARS-CoV-2 at a high multiplicity of infection (MOI) of 2, but pretreatment with camostat efficiently blocked virus infection as evidenced by confocal microscopy on hAOs fixed at 16 hr postinfection (*Figure 5A*). Cathepsin inhibition did not affect entry. At 24 hr

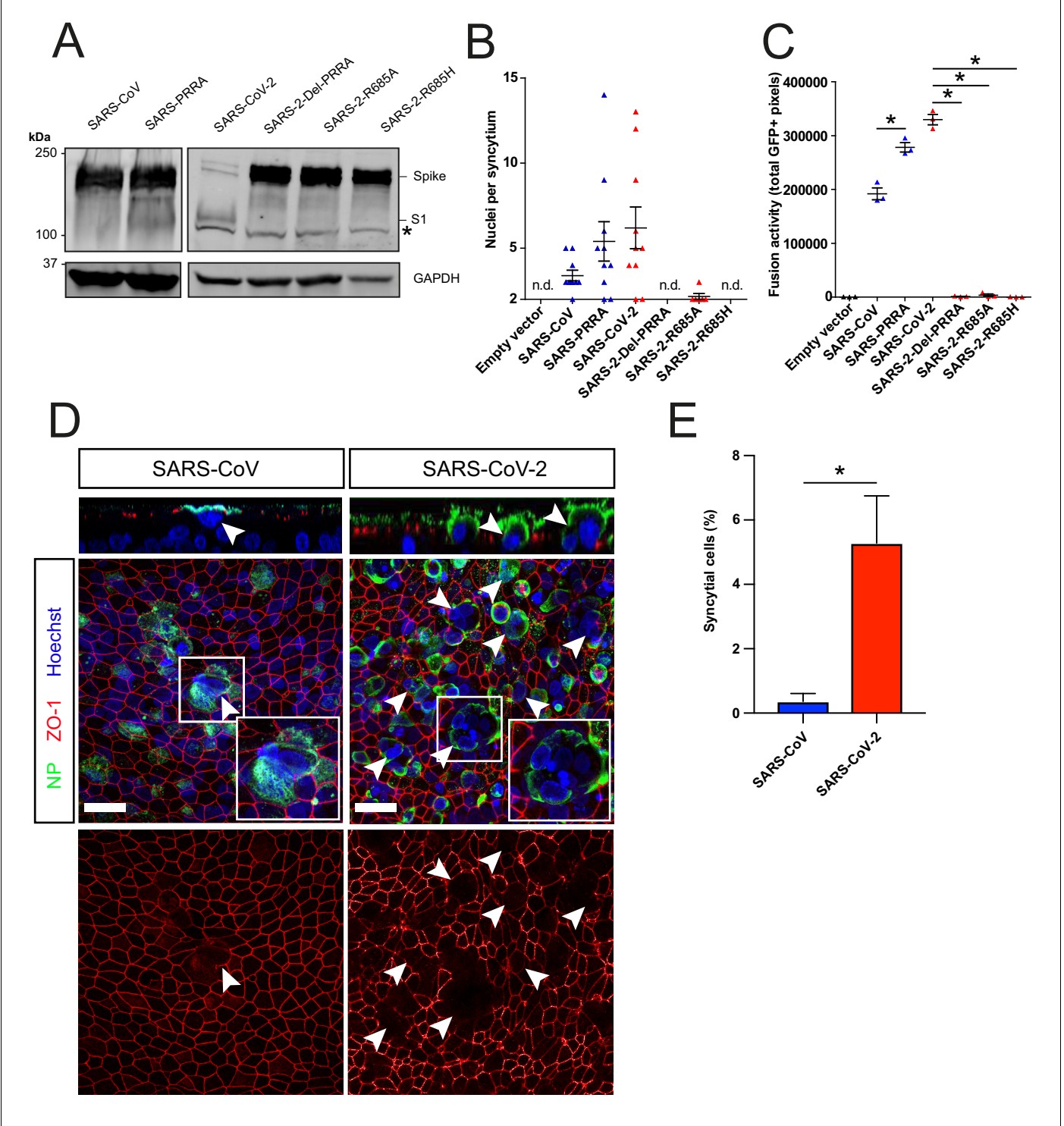

**Figure 3.** The SARS-CoV-2 multibasic cleavage site facilitates cell–cell fusion and SARS-CoV-2 is more fusogenic than SARS-CoV on human airway organoids. (**A**) Proteolytic cleavage of SARS-CoV-2 S, SARS-CoV S, and S mutants was assessed by overexpression in HEK-293T cells and subsequent western blots for S1. GAPDH was used as a loading control. Asterisk indicates an unspecific band. (**B and C**) Fusogenicity of SARS-CoV-2 S, SARS-CoV S, and S mutants was assessed after 18 hr by counting the number of nuclei per syncytium (**B**) and by measuring the sum of all GFP+ pixels per well (**C**). Statistical analysis was performed by one-way ANOVA on SARS-CoV or SARS-CoV-2 S-mediated fusion compared with its respective mutants. *$p<0.05$ (**C**). (**D**) Differentiated bronchiolar hAO cultures were infected at an MOI of 1 with SARS-CoV or SARS-CoV-2. Seventy-two hours postinfection they were fixed and stained for nucleoprotein (NP; green) and tight junctions (ZO1; red) to image syncytia. Nuclei were stained with hoechst (blue). Scale bars indicate 20 μm. Arrows indicate syncytial cells. (**E**) Percentage of syncytial cells of total number of infected cells per field of 0.1 mm². Five fields were

*Figure 3 continued on next page*

*Figure 3 continued*

counted. T-test was performed for statistical analysis. *p<0.05. Experiments were performed in triplicate (**C**). Representative experiments from at least two independent experiments are shown. Error bars indicate SEM. hAO, human airway organoids; H p.i., hours postinfection; MOI, multiplicity of infection; n.d., not detected.

The online version of this article includes the following figure supplement(s) for figure 3:

**Figure supplement 1.** A GFP complementation based assay for assessing coronavirus fusogenicity.

postinfection, SARS-CoV-2 infection spread in hAOs treated with dimethyl sulfoxide (DMSO) or E64D, but only rare single cells were observed after camostat treatment (*Figure 5B*). Next, we tested whether virus replication was affected by camostat pretreatment of the hAOs. After infection at an MOI of 2, replication was assessed at 2, 24, and 48 hr postinfection by RT-qPCR and live virus titration. In the control hAOs, SARS-CoV-2 replicated to high titers, while camostat reduced replication by approximately 90% (*Figure 5C–E*). We also tested the effect of camostat in 2D differentiated hAOs at air–liquid interface using a low MOI of 0.1. Here, viral titers in apical washes did not increase after camostat pretreatment (*Figure 5F*), whereas replication to moderate titers was observed in the control wells. These findings indicate that SARS-CoV-2 utilizes serine proteases for efficient entry into relevant human airway cells and serine protease inhibition decreases replication.

## Discussion

SARS-CoV-2 harbors an MBCS in its S protein. Recent findings show that replacing this site with the SARS-CoV monobasic cleavage site decreases PP infectivity on the adenocarcinoma cell line Calu-3, suggesting that this motif is a human airway adaptation (*Hoffmann et al., 2020a*). This raised the question whether similar findings would be obtained in relevant lung cells. In this study, we found that the SARS-CoV-2 MBCS alters tropism by increasing infectivity on hAOs. Furthermore, we report that the MBCS increases S protein fusogenicity, entry rate, and serine protease usage. Blocking serine proteases, but not cathepsins, in hAOs effectively inhibited SARS-CoV-2 entry and replication, suggesting that serine protease-mediated entry is the main entry route in vivo.

In contrast to SARS-CoV-2, SARS-CoV does not contain an MBCS, yet infects Calu-3 cells with similar efficiency. Introducing an MBCS to SARS-CoV S did not increase infectivity, indicating that the SARS-CoV S has other adaptations to enter airway cells. These data are in agreement with a study that observed no benefit of furin cleavage on SARS-CoV infectivity (*Follis et al., 2006*). Whereas SARS-CoV-2 appears to have adapted to increase fusogenicity and serine protease-mediated S activation for rapid plasma membrane entry, SARS-CoV may have specific adaptations to enter these cells more slowly. Slower viral dissemination may explain why most SARS-CoV patients entered the infectious phase of the disease after symptom onset (*Cheng et al., 2004*; *Peiris et al., 2003a*). This could have played a role in the 2003 SARS-CoV epidemic, allowing strict public health interventions including quarantining of symptomatic people and contact tracing to halt viral spread. For SARS-CoV-2, however, several studies have reported that individuals can transmit the virus to others before they become symptomatic (*Bai et al., 2020*; *Huang et al., 2020*; *Kimball et al., 2020*; *Wei et al., 2020b*; *Wiersinga et al., 2020*). Whether differences in entry rate allow SARS-CoV-2 to spread more efficiently in the human airway compared with SARS-CoV remains to be investigated. It will be interesting to assess this using authentic SARS-CoV-2 containing MBCS mutations, which requires a reverse genetic system, not available to this study at present. Whether cell–cell fusion also plays a role in virus dissemination needs to be determined. In a cell–cell fusion assay and in hAOs cultured at air–liquid interface, we show that SARS-CoV-2 is more fusogenic than SARS-CoV and that fusogenicity is increased by the SARS-CoV-2 S MBCS. The role of cell–cell fusion in coronavirus transmission and pathogenesis has not been investigated in detail, but it could be a strategy to avoid extracellular immune surveillance and may increase the viral dissemination rate in the airways in vivo. Whether the MBCS also affects entry into cells of other organs needs to be investigated further. Although SARS-CoV-2 symptoms are mainly respiratory, recent reports indicate frequent extrapulmonary manifestations, including but not limited to thrombotic complications, acute kidney injury, gastrointestinal symptoms, dermatologic complications, and anosmia (*Gupta et al., 2020*). Of note, acute kidney injury was uncommon during the SARS-CoV epidemic (*Chu et al., 2005*). It is

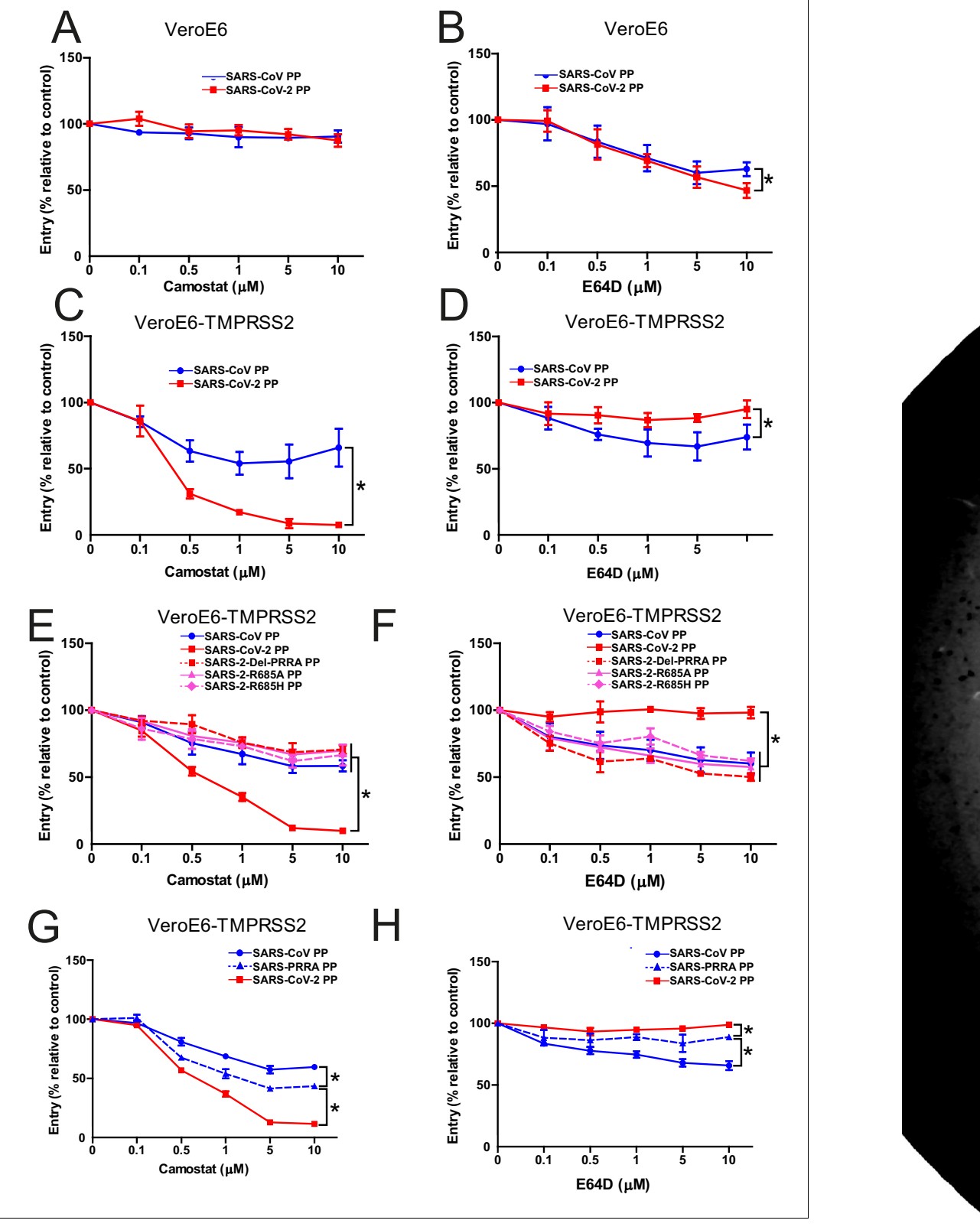

**Figure 4.** The SARS-CoV-2 multibasic cleavage site increases serine protease usage. (**A and B**) SARS-CoV PP and SARS-CoV-2 PP entry route on cells pretreated with a concentration range of camostat (**A**) or E64D (**B**) to inhibit serine proteases and cathepsins, respectively. (**C and D**) SARS-CoV and SARS-CoV-2 PP entry route on VeroE6-TMPRSS2 cells pretreated with a concentration range of camostat (**C**) or E64D (**D**) to inhibit serine proteases and cathepsins, respectively. T-test was performed for statistical analysis at the highest concentration. *p<0.05. (**E and F**) Entry route of SARS-CoV-2 PP

*Figure 4 continued on next page*

*Figure 4 continued*

and MBCS mutants on VeroE6-TMPRSS2 cells pretreated with a concentration range of camostat (E) or E64D (F) to inhibit serine proteases and cathepsins, respectively. One-way ANOVA was performed for statistical analysis comparing all groups to SARS-CoV-2 PPs at the highest concentration. *p<0.05. (G and H) Entry route of SARS-CoV PPs and SARS-PRRA PPs on VeroE6-TMPRSS2 cells pretreated with a concentration range of camostat (G) or E64D (H) to inhibit serine proteases and cathepsins, respectively. One-way ANOVA was performed for statistical analysis comparing all groups to SARS-PRRA PPs at the highest concentration. *p<0.05. ANOVA, analysis of variance; MBCS, multibasic cleavage site; PP, pseudoparticles. Representative experiments in triplicate from at least two independent experiments are shown. Error bars indicate SD.

unclear at this moment whether these manifestations are the result of extrapulmonary viral replication.

Using VeroE6-TMPRSS2 cells that have both active serine protease- and cathepsin-mediated entry pathways, we show that the MBCS increases serine protease-mediated S activation, while decreasing cathepsin-mediated S activation. This indicates that the MBCS could be an adaptation to serine protease-mediated entry. Whether this site improves S activation by any protease or by serine proteases specifically remains to be tested. Encountering serine proteases first may result in more plasma membrane entry over endosomal entry. More efficient fusion of multibasic motif containing S proteins may be caused by increased S2′ cleavage due to higher accessibility of a S1/S2 cleaved S compared with an uncleaved S. S1/S2 cleavage was recently shown to increase the binding of S to ACE2 (*Wrobel et al., 2020*). Recent reports indicate that neuropilins (NRP) may be an additional entry factor for SARS-CoV-2 (*Cantuti-Castelvetri et al., 2020*; *Daly et al., 2020*). This is an interesting hypothesis as ACE2 expression (mRNA and protein) is relatively low in the lungs, whereas NRP expression appears to be abundant. NRP1 binds peptides with an exposed C-terminal arginine, such as furin-cleaved proteins, and a SARS-CoV-2 mutant with an altered MBCS did not depend on NRP1 for infectivity (*Cantuti-Castelvetri et al., 2020*). Therefore, it was concluded that NRP1 may promote the interaction of the S1/S2-cleaved virus with ACE2. However, Daly and colleagues observed that NRP1 knockdown did not affect cellular attachment, but did enhance entry and syncytium formation. Therefore, we suggest that NRP1 may facilitate S2′ cleavage by exposing this site or S1 dissociation after S2′ cleavage for timely exposure of the fusion peptide and subsequent fusion. Structural changes caused by S1/S2 cleavage may affect protease accessibility or protein interactions at the cell membrane as well as increase subsequent S2′ cleavage.

While mutations in the SARS-CoV-2 MBCS decreased airway cell infectivity, they increased infectivity on VeroE6 cells. Several groups have reported mutations or deletions in or around the SARS-CoV-2 MBCS that arise in cell culture on VeroE6 cells (*Klimstra et al., 2020*; *Lau et al., 2020*; *Ogando et al., 2020*), indicating that the lack of an MBCS creates a selective advantage in cell culture on VeroE6 cells. The mechanism behind this remains unknown. The increased infectivity of MBCS mutants was not observed by *Hoffmann et al., 2020a*, but in that study, complete cleavage motifs including four amino acids N-terminally from the minimal RXXR cleavage site were exchanged between SARS-CoV-2 and SARS-CoV (*Hoffmann et al., 2020a*). In contrast, we mutated single sites or removed/inserted only the PRRA motif. Importantly, a cell culture-adapted virus containing a complete deletion of the MBCS was recently shown to be attenuated in hamsters (*Lau et al., 2020*). These studies support our findings that the SARS-CoV-2 MBCS affects tropism, facilitates airway cell entry, and show that proper characterization of virus stocks is essential.

Entry inhibition has been proposed as an effective treatment option for SARS-CoV-2. Chloroquine can block SARS-CoV and SARS-CoV-2 entry in vitro into VeroE6 cells (*Vincent et al., 2005*; *Wang et al., 2020*), but does not block entry into cells expressing serine proteases (Calu-3 and Vero-TMPRSS2) (*Hoffmann et al., 2020c*). This is expected, as chloroquine acts in the endosome, while the endosomal entry pathway is not utilized in serine protease expressing cells. As lung cells express serine proteases, inhibitors that block endosomal entry are likely to be ineffective in vivo. These findings highlight that drug screens should be performed directly in relevant cells to prevent wasting resources. Our study shows that SARS-CoV-2 replication in hAOs infected with a high MOI is inhibited ~90% by camostat, suggesting that this drug may be effective in vivo. Future studies assessing the efficacy and safety of camostat in animal models should be conducted. For SARS-CoV, camostat improved survival to 60% in a lethal mouse model (*Zhou et al., 2015*). In the same study, inhibition of cathepsins using a cysteine protease inhibitor was ineffective, supporting a critical role for serine proteases in viral spread and pathogenesis in vivo. In Japan, camostat has been clinically

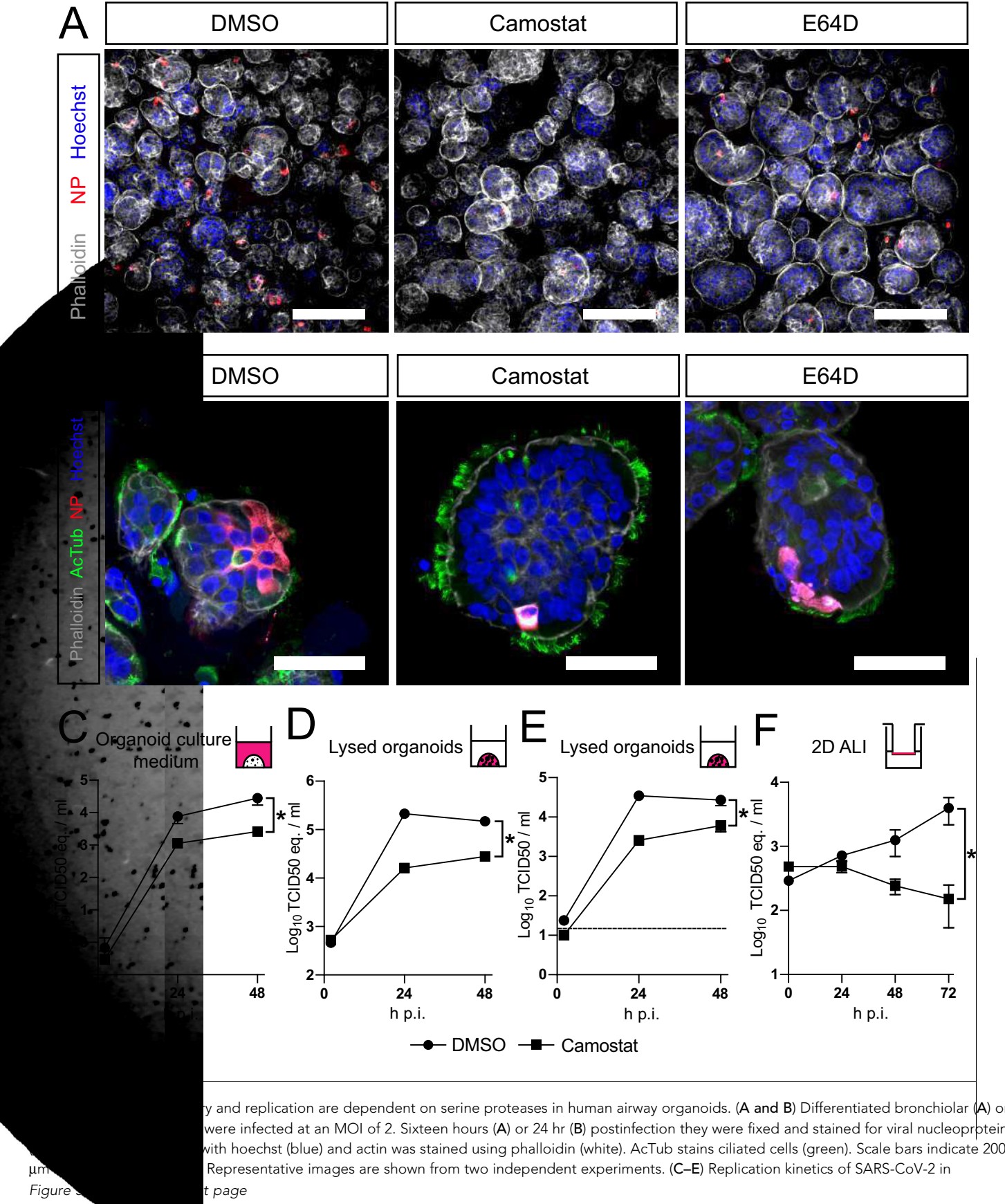

and replication are dependent on serine proteases in human airway organoids. (A and B) Differentiated bronchiolar (A) or
were infected at an MOI of 2. Sixteen hours (A) or 24 hr (B) postinfection they were fixed and stained for viral nucleoprotein
with hoechst (blue) and actin was stained using phalloidin (white). AcTub stains ciliated cells (green). Scale bars indicate 200
µm Representative images are shown from two independent experiments. (C–E) Replication kinetics of SARS-CoV-2 in

*Figure ... next page*

*Figure 5 continued*

bronchiolar hAO cultures pretreated with camostat or carrier (DMSO). (**C and D**) TCID50 equivalents (eq.) per mL are shown in culture medium (**C**) and lysed organoids (**D**). Circles indicate DMSO-treated organoids, whereas squares indicate camostat-treated organoids. (**E**) Live virus titers (TCID50/mL) in lysed organoids. Dotted line indicates limit of detection. (**F**) Replication kinetics of SARS-CoV-2 in 2D tracheal air–liquid interface airway cultures pretreated with camostat or carrier (DMSO). TCID50 eq./mL in apical washes are shown. Two-way ANOVA was performed for statistical analysis. Error bars indicate SEM. *$p<0.05$. DMSO, dimethyl sulfoxide; hAO, human airway organoid; H p.i., hours postinfection; MOI, multiplicity of infection.

approved to treat chronic pancreatitis and thus represents a potential therapy for respiratory coronavirus infections.

We demonstrate that SARS-CoV-2 enters hAOs using serine proteases, but not using cathepsins, clarifying existing discrepancies between human cell lines and suggesting that serine-protease-mediated entry is the main entry route in vivo. In addition, our findings indicate that the multibasic cleavage motif in the SARS-CoV-2 S protein is an adaptation to this viral entry strategy, and can guide the design of entry inhibitors to combat COVID-19.

# Materials and methods

**Key resources table**

| Reagent type (species) or resource | Designation | Source or reference | Identifiers | Additional information |
|---|---|---|---|---|
| VeroE6 cells (*Cercopithecus aethiops*) | Monkey kidney cell line | ATCC | CRL 1586TM | |
| Vero cells (*Cercopithecus aethiops*) | Monkey kidney cell line | WHO | RCB 10-87 | |
| Calu-3 (*Homo sapiens*) | Lung adenocarcinoma cell line | ATCC | HTB 55 | |
| SARS-CoV-2 BavPat1 | SARS-CoV-2 | Dr. Christian Drosten | European Virus Archive Global #026V-03883 | |
| SARS-CoV HKU39849 | SARS-CoV | Dr. Malik Peiris | N/A | |
| Airway tissue for organoids (*Homo sapiens*) | Airway organoids | This study | This study | |
| Aloxistatin | E64D | MedChemExpress | Cat# HY-100229 | |
| Camostat mesylate | Camostat | Sigma | Cat# SML0057 | |
| Hexadimethrine bromide | Polybrene | Sigma | 107689-10G | |
| Polyethylenimine linear | Polyethylenimine | Polysciences | Cat# 23966 | |
| Hygromycin B | Hygromycin B | Invitrogen | Cat# 10843555001 | |
| G418, Geneticin | Geneticin | Invitrogen | Cat# 10131035 | |
| Opti-MEM I (1×) + GlutaMAX | Opti-MEM I (1×) + GlutaMAX | Gibco | Cat# 51985-042 | |
| Advanced DMEM/F12 | Advanced DMEM/F12 | Thermo Fisher scientific | Cat# 12634-010 | |
| AO medium | AO medium | *Sachs et al., 2019* | N/A | |
| Pneumacult ALI medium | Pneumacult ALI medium | Stemcell | Cat # 05001 | |

*Continued on next page*

*Continued*

| Reagent type (species) or resource | Designation | Source or reference | Identifiers | Additional information |
|---|---|---|---|---|
| TrypIE | TrypIE | Thermo Fisher Scientific | Cat# 12605010 | |
| Cultrex Basement Membrane Extract, Type 2 | Basement membrane extract | R&D Systems | Cat# 3533-005-02 | |
| 12 mm Transwell with 0.4 µm Pore Polyester Membrane Insert, Sterile | Transwell inserts | Corning | Cat# 3460 | |
| Collagen Type I, High concentration Rat tail | Collagen | Corning | Cat# 354249 | |
| 0.45 µm low protein binding filter | 0.45 µm low protein binding filter | Millipore | Cat# SLHV033RS | |
| Recombinant DNA reagent | pCMV-S (CUHK-W1) | Sino Biological | Cat# VG40150-G-N | Encoding S of isolate CUHK-W1; |
| Recombinant DNA reagent | pCAGGS-S (CUHK-W1) | This study | This study | Encoding S of isolate CUHK-W1; |
| Recombinant DNA reagent | pCAGGS-S (Wuhan-Hu-1) | This study | This study | Encoding S of isolate Wuhan-Hu-1 |
| Recombinant DNA reagent | pQXCIN | Clontech | Cat# 631516 | Retro-X Q vector set |
| Recombinant DNA reagent | pQXCIH | Clontech | Cat# 631516 | Retro-X Q vector set |
| Recombinant DNA reagent | pQXCIP-GFP1-10 | Addgene | Cat# 68715 | GFP1-10 |
| Recombinant DNA reagent | pCDNA-TMPRSS2-FLAG | Genscript | OHu13675D | Human TMPRSS2 cDNA |
| Recombinant DNA reagent | pCAGGS-β-Actin-7xGFP11-P2A-BFP | This study | This study | β-Actin-7xGFP11-P2A-BFP |
| Recombinant DNA reagent | pBS-gag-pol | Addgene | Cat# 35614 | gag-pol |
| Recombinant DNA reagent | pMD2.G | Addgene | Cat# 12259 | VSV-G |
| Recombinant DNA reagent | pVSV-eGFP-dG | Addgene | Cat# 31842 | VSV delta G genomic plasmid |
| Recombinant DNA reagent | pCAG-VSV-P | Addgene | Cat# 64088 | P protein |
| Recombinant DNA reagent | pCAG-VSV-L | Addgene | Cat# 64085 | L protein |
| Recombinant DNA reagent | pCAG-VSV-N | Addgene | Cat# 64087 | N protein |
| Recombinant DNA reagent | pCAGGS-T7Opt | Addgene | Cat# 65974 | T7 polymerase |
| Antibody | Mouse-anti-SARS-CoV NP (monoclonal) | Sino Biological | Cat# 40143-MM05 | IF (1:400) |
| Antibody | Rabbit-anti-SARS-CoV NP (polyclonal) | Sino Biological | Cat# 40143-T62 | IF (1:400) |

*Continued on next page*

*Continued*

| Reagent type (species) or resource | Designation | Source or reference | Identifiers | Additional information |
|---|---|---|---|---|
| Antibody | Goat anti-ACE2 (polyclonal) | R&D Systems | Cat# AF933 | IF (1:200) |
| Antibody | Mouse anti-TMPRSS2 (monoclonal) | Santa Cruz | Cat# sc-515727 | IF (1:200) |
| Antibody | Rabbit-anti-goat | Dako | Cat# P0160 | IF (1:400) |
| Antibody | Goat-anti-mouse | Dako | Cat# P0260 | IF (1:400) |
| Antibody | Mouse anti-VSV (monoclonal) | Absolute Antibody | Cat# EB0010 | Pseudoparticle production (1:50,000) |
| Antibody | Goat anti-rabbit IgG (H+L) Alexa Fluor Plus 488 | Invitrogen | Cat# A32731 | IF (1:400) |
| Antibody | Goat anti-mouse IgG (H+L) Alexa Fluor Plus 488 | Invitrogen | Cat# A11029 | IF (1:400) |
| Antibody | Goat anti-mouse IgG (H+L) Alexa Fluor Plus 594 | Invitrogen | Cat# A21125 | IF (1:400) |
| Antibody | Mouse anti-double stranded RNA IgG2A (monoclonal) | Scicons | J2 clone | IF (1:500) |
| Antibody | Mouse-anti-ZO1 IgG1 (monoclonal) | Invitrogen | Cat# 33-9100 | IF (1:200) |
| Antibody | Mouse-anti-CC10 IgG1 Alexa Fluor 594 (monoclonal) | Santa Cruz Biotechnology | Cat# sc-390313 AF594 | IF (1:100) |
| Antibody | Mouse-anti-AcTub IgG2A Alexa Fluor 488 (monoclonal) | Santa Cruz Biotechnology | Cat# sc-23950 AF488 | IF (1:100) |
| Antibody | Mouse anti-MUC5AC (monoclonal) | Invitrogen | Cat# MA5-12178 | IF (1:100) |
| Antibody | Mouse-anti-FOXJ1 IgG1 (monoclonal) | eBioscience | Cat# 14-9965-82 | IF (1:200) |
| Antibody | Rabbit anti-SARS-CoV S1 (polyclonal) | Sino Biological | Cat# 40150-T62 | WB (1:1000) |
| Antibody | Mouse anti-GAPDH (monoclonal) | Santa Cruz Biotechnology | Cat# sc-32233 | WB (1:1000) |
| Pierce Silver Stain for Mass Spectrometry | Silver stain | Pierce | Cat# 24600 | |
| TO-PRO-3 Iodide | TO-PRO3 | ThermoFisher | Cat# T3605 | |
| Phalloidin CruzFluor 647 Conjugate | Phalloidin | Santa Cruz | Cat# ec-363796 | |
| Hoechst 33342, Trihydrochloride, Trihydrate | Hoechst | ThermoFisher | Cat# H1399 | |

*Continued on next page*

*Continued*

| Reagent type (species) or resource | Designation | Source or reference | Identifiers | Additional information |
|---|---|---|---|---|
| 4× Laemmli Sample Buffer | Laemmli | Bio-Rad | Cat# 1610747 | |
| Odyssey CLx | Odyssey CLx | Licor | | |
| Amersham Typhoon Biomolecular Imager | Amersham Typhoon Biomolecular Image | GE Healthcare | | |
| Amersham Imager 600 | Amersham Imager 600 | GE Healthcare | | |
| LSM700 confocal microscope | LSM700 confocal microscope | Zeiss | | |
| Carl ZEISS Vert.A1 | Carl ZEISS Vert.A1 | Zeiss | | |
| ZEN software | ZEN | Zeiss | | |
| ImageQuant TL 8.2 | ImageQuant TL 8.2 | GE Healthcare | | |
| Studio Lite Ver 5.2 | Studio Lite Ver 5.2 | Licor | | |
| GraphPad PRISM 8 | GraphPad PRISM 8 | GraphPad | | |
| Adobe Illustrator | Illustrator | Adobe Inc | | |

## Viruses and cells

Vero, VeroE6, and VeroE6 stable cell lines were maintained in Dulbecco's modified Eagle's medium (DMEM, Gibco) supplemented with 10% fetal calf serum (FCS), HEPES (20 mM, Lonza), sodium bicarbonate (0.075%, Gibco), penicillin (100 IU/mL), and streptomycin (100 IU/mL) at 37°C in a humidified $CO_2$ incubator. Calu-3 and Calu-3-stable cell lines were maintained in Eagle's minimal essential medium (ATCC) supplemented with 20% FCS, penicillin (100 IU/mL), and streptomycin (100 IU/mL) at 37°C in a humidified $CO_2$ incubator. HEK-293T cells were cultured in DMEM supplemented with 10% FCS, sodium pyruvate (1 mM, Gibco), non-essential amino acids (1×, Lonza), penicillin (100 IU/mL), and streptomycin (100 IU/mL) at 37°C in a humidified $CO_2$ incubator. TMPRSS2 and GFP1-10 overexpression cells were maintained in medium containing hygromycin (Invitrogen) and geneticin (Invitrogen), respectively. SARS-CoV-2 (isolate BetaCoV/Munich/BavPat1/2020; European Virus Archive Global #026 V-03883; kindly provided by Dr. C. Drosten) and SARS-CoV (isolate HKU39849) were propagated on Vero cells in Opti-MEM I (1×) + GlutaMAX (Gibco), supplemented with penicillin (100 IU/mL) and streptomycin (100 IU/mL) at 37°C in a humidified $CO_2$ incubator. Vero, VeroE6, and Calu-3 cells were purchased from ATCC. Cultures were routinely tested for mycoplasma. The SARS-CoV-2 isolate was obtained from a clinical case in Germany, diagnosed after returning from China. Stocks were produced by infecting cells at an MOI of 0.01 and incubating the cells for 72 hr. The culture supernatant was cleared by centrifugation and stored in aliquots at −80°C. Stock titers were determined by preparing 10-fold serial dilutions in Opti-MEM I (1×) + GlutaMAX. Aliquots of each dilution were added to monolayers of $2 \times 10^4$ VeroE6 cells in the same medium in a 96-well plate. Plates were incubated at 37°C 5% $CO_2$ for 5 days and then examined for cytopathic effect. The TCID50 was calculated according to the method of Spearman and Kärber. All work with infectious SARS-CoV and SARS-CoV-2 was performed in a Class II Biosafety Cabinet under BSL-3 conditions at Erasmus Medical Center.

## Cloning

Codon-optimized SARS-CoV (isolate CUHK-W1; VG40150-G-N) S expression plasmids (pCMV) were ordered from Sino-Biological and subcloned into pCAGGS using the ClaI and KpnI sites. The last 19

amino acids containing the Golgi retention signal of the SARS-CoV S protein were deleted to enhance PP production. Codon-optimized cDNA encoding SARS-CoV-2 S glycoprotein (isolate Wuhan-Hu-1) with a C-terminal 19 amino acid deletion was synthesized and cloned into pCAGSS in between the EcoRI and BglII sites. S expressing pCAGGS vectors were used for GFP complementation fusion assays, and equivalent S proteins with the C-terminal deletion were used for the production of PPs, as described in Material and methods. The cDNA encoding GFP1-10 was obtained from Addgene and was subcloned into pQXCIN (Clontech) in between BamHI and EcoRI to obtain the pQXCIN-GFP1-10 vector. The cDNA encoding human TMPRSS2 (NM_005656; OHu13675D) was obtained from Genscript. The cDNA fused to a C-terminal HA tag was subcloned into pQXCIH (Clontech) in between the NotI and PacI sites to obtain the pQXCIH-TMPRRS2-HA vector. A synthetic DNA construct of β-actin-7xGFP11-P2A-BFP was ordered from GenScript and subcloned into pGAGGS using EcoRI and BglII. SARS-CoV and SARS-CoV-2 S protein mutations (SARS-PRRA, SARS-2-Del-PRRA, SARS-2-R685A, SARS-2-R685H) were generated by subcloning synthetic DNA constructs (Genscript) containing the desired mutations into the pCAGGS-S vectors or by mutagenesis PCR.

## Isolation, culture, and differentiation of human airway stem cells

Adult lung tissue was obtained from residual, tumor-free material obtained at lung resection surgery for lung cancer. The Medical Ethical Committee of the Erasmus MC Rotterdam granted permission for this study (METC 2012–512). Isolation, culture, and differentiation were performed as described previously (*Lamers et al., 2020*) according to a protocol adapted from *Sachs et al., 2019*. Differentiation time on air–liquid interphase was 10–11 weeks. For this study, we used carefully dissected out bronchial material for the generation of human bronchial airway organoids. Bronchiolar organoids were generated from distal lung parenchymal material. Tracheal stem cells were collected from tracheal aspirates of intubated preterm infants (<28 weeks gestational age) (*Hiemstra et al., 2020*) and cultured as described before (*Sachs et al., 2019*). For the tracheal aspirates, informed consent was obtained from parents, and approval was given by the Medical Ethical Committee (METC no. MEC-2017–302). All donor materials were completely anonymized.

## Authentic virus infection of primary airway cells

To assess differences in syncytium formation, 2D air–liquid interface differentiated airway cultures were washed three times with 500 µL advanced DMEM/F12 (AdDF+++, Gibco) before inoculation from the apical side at an MOI of 1 in 200 µL AdDF+++ per well. Next, cultures were incubated at 37°C 5% $CO_2$ for 2 hr before washing four times in 500 µL AdDF+++. Cultures were washed daily from the apical side with 300 µL AdDF+++ to facilitate virus spread. At 72 hr postinfection, cells were fixed for immunofluorescent staining.

To determine the effect of camostat on SARS-CoV-2 entry, we incubated bronchial or bronchiolar cultures that were differentiated at air–liquid interface for 10–11 weeks with 100% dispase in the basal compartment of a 12 mm Transwell insert. After a 10 min incubation step at 37°C 5% $CO_2$, dispase was removed and cold 500 µL AdDF+++ was pipetted onto the apical side of the Transwell to dislodge the pseudostratified epithelial layer, which was subsequently mechanically sheared by pipetting using a P1000 tip. The resulting epithelial fragments were washed twice in 5 mL AdDF+++ before treatment with 10 µM camostat, 10 µM E64D, or carrier (DMSO) in Pneumacult (PC) ALI medium (Stemcell) on ice for 1 hr. Next, fragments were infected at an MOI of 2 for 2 hr at 37°C 5% $CO_2$ in the presence of inhibitors or DMSO. Subsequently, fragments were washed three times in 5 mL cold AdDF+++ before being embedded in 30 µL BME (Type 2; R and D Systems) per well in a 48-well plate. Approximately 200,000 cells were plated per well. After solidification of the BME, 200 µL PC was added per well and plates were incubated at 37°C 5% $CO_2$.

To assess SARS-CoV-2 replication in the presence of camostat, bronchiolar airway organoids, or 2D air–liquid interface differentiated tracheal airway cultures were infected as described above. For organoids, culture medium was collected at the indicated time points and frozen at −80°C. After culture medium collection, BME droplets containing organoids were resuspended in 200 µL AdDF+++ and samples were frozen at −80°C to lyse the cells. To assess 2D air–liquid interface differentiated airway culture replication kinetics, apical washes were collected at the indicated time points by adding 200 µL AdDF+++ apically, incubating for 15 min at 37°C 5% $CO_2$, and collecting the sample

before storage at −80℃. For virus titrations using RT-qPCR (*Lamers et al., 2020*) or TCID50 determination, samples were thawed and centrifuged at 500 × g for 3 min. For TCID50 determination, six replicates were performed per sample.

## Fixed immunofluorescence microscopy and immunohistochemistry

Transwell inserts were fixed in formalin, permeabilized in 70% ethanol, and blocked for 60 min in 10% normal goat serum or 3% bovine serum albumin in phosphate-buffered saline (PBS) (blocking buffer). For organoids, 0.1% triton X-100 was added to the blocking buffer to increase antibody penetration. Cells were incubated with primary antibodies overnight at 4℃ in blocking buffer, washed twice with PBS, incubated with corresponding secondary antibodies Alexa488- and 594-conjugated secondary antibodies (1:400; Invitrogen) in blocking buffer for 2 hr at room temperature, washed two times with PBS, incubated with indicated additional stains (TO-PRO3, phalloidin-633 [SC-363796, Santa Cruz Biotechnology], Hoechst), washed twice with PBS, and mounted in Prolong Antifade (Invitrogen) mounting medium.

SARS-CoV-2 and SARS-CoV were stained with mouse-anti-SARS-CoV nucleoprotein (40143-MM05, 1:400, Sino Biological) or rabbit-anti-SARS-CoV nucleoprotein (40143-T62, 1:400, Sino Biological). Tight junctions were stained using mouse-anti-ZO1 (ZO1-1A12, 1:200, Invitrogen). Club cells and goblet cells were stained with mouse-anti-CC10 (sc-390313 AF594, 1:100, Santa Cruz Biotechnology) and mouse anti-MUC5AC (MA5-12178, 1:100, Invitrogen), respectively. Ciliated cells were stained with mouse-anti-FOXJ1 (14-9965-82, 1:200, eBioscience) and mouse-anti-AcTub (sc-23950 AF488, 1:100, Santa Cruz Biotechnology). For lineage marker stainings, formalin-fixed inserts were paraffin-embedded, sectioned, and deparaffinized as described before prior to staining (*Rockx et al., 2020*). Samples were imaged on an LSM700 confocal microscope using ZEN software (Zeiss). Representative images were acquired and shown as Z-projections, single slices, or XZ cross sections.

Immunohistochemistry was performed as described previously (*Rockx et al., 2020*) on formalin-fixed, paraffin-embedded Transwell inserts. ACE2 and TMPRSS2 were stained using goat-anti-hACE2 (AF933, 1:200, R and D Systems) and mouse-anti-TMPRSS2 (sc-515727, 1:200, Santa Cruz Biotechnology) and visualized with rabbit-anti-goat (P0160, 1:200, Dako) and goat-anti-mouse (P0260, 1:100, Dako) horseradish peroxidase-labeled secondary antibody, respectively. Samples were counterstained using hematoxylin.

## Generation of stable cell lines expressing GFP1-10 and TMPRSS2

VeroE6 GFP1-10, VeroE6-TMPRSS2 cells, VeroE6-TMPRSS2 GFP1-10 cells, and Calu-3 GFP1-10 cells were generated by retroviral transduction. To produce the retrovirus, 10 µg pQXCIH-TMPRRS2-HA or pQXCIN-GFP1-10 was co-transfected with polyethylenimine (PEI) with 6.5 µg pBS-gag-pol (Addgene #35614) and 5 µg pMD2.G (Addgene #12259) in a 10 cm dish of 70% confluent HEK-293T cells in Opti-MEM I (1×) + GlutaMAX. Retroviral particles were harvested at 72 hr post-transfection, cleared by centrifugation at 2000 × g, filtered through a 0.45 µm low protein binding filter (Millipore), and used to transduce designated cells. Polybrene (Sigma) was added at a concentration of 4 µg/mL to enhance transduction efficiency. Transduced cells were selected with hygromycin B (Invitrogen) for TMPRSS2 cells and/or geneticin (Invitrogen) for GFP1-10 cells.

## GFP complementation fusion assay

HEK-293T cells were grown in six-well format to 70–80% confluency and were transfected with 1.5 µg pGAGGS-spike (all coronavirus S variants described above) DNA and pGAGGS-β-Actin-P2A-7xGFP11-BFP DNA or empty vector DNA with PEI in a ratio of 1:3 (DNA:PEI). Beta-actin was tagged with 7xGFP11 expressed in tandem and blue fluorescent protein (BFP). The two genes were separated by a P2A self-cleaving peptide. Two variants of this construct were used. One variant contained a GSG linker located N-terminally from the P2A site to improve self-cleavage and this construct was used in qualitative confocal microscopy experiments. A variant lacking the GSG linker was less efficiently cleaved as indicated by both cytoplasmic and nuclear localized GFP, but this generated an equal distribution of GFP throughout the cell and therefore it was used for all fusion assays in which the sum of all GFP+ pixels was calculated. Transfected HEK-293T cells were incubated overnight at 37℃ 5% $CO_2$. GFP1-10 expressing cells were seeded in a 12-well plate to achieve 90–100%

confluency after overnight incubation at 37°C 5% $CO_2$ and medium was refreshed with Opti-MEM I (1×) + GlutaMAX. HEK-293T cells were resuspended in PBS by pipetting to generate a single-cell suspension and added to GFP1-10 expressing cells in a ratio of 1:80 (HEK-293T cells: GFP1-10 expressing cells). Fusion events were quantified by detecting GFP+ pixels after 18 hr incubation at 37°C 5% $CO_2$ using Amersham Typhoon Biomolecular Imager (channel Cy2; resolution 10 µm; GE Healthcare). Data was analyzed using the ImageQuant TL 8.2 image analysis software (GE Healthcare) by calculating the sum of all GFP+ pixels per well. For nuclear counting fluorescence microscopy images were obtained with a Carl ZEISS Vert.A1 microscope paired with an AxioCam ICm1 camera and Colibri seven laser (469/38 nm for GFP and 365/10 nm for BFP) using ZEN analysis software (20× magnification). Nuclei per syncytia were calculated by counting BFP-positive nuclei after 18 hr incubation at 37°C 5% $CO_2$. Confocal microscopy images were taken on an LSM700 confocal microscope using ZEN software. Representative images were acquired and shown as single slices.

### VSV delta G rescue

The protocol for VSV-G PP rescue was adapted from *Whelan et al., 1995*. VSV rescue plasmids pVSV-eGFP-dG (#31842), pMD2.G (#12259), pCAG-VSV-P (#64088), pCAG-VSV-L (#64085), pCAG-VSV-N (#64087), and pCAGGS-T7Opt (#65974) were ordered from Addgene. Briefly, a 70% confluent 10 cm dish of HEK-293T cells was transfected with 10 µg pVSV-eGFP-dG, 2 µg pCAG-VSV-N (nucleocapsid), 2 µg pCAG-VSV-L (polymerase), 2 µg pMD2.G (glycoprotein, VSV-G), 2 µg pCAG-VSV-P (phosphoprotein), and 2 µg pCAGGS-T7Opt (T7 RNA polymerase) using PEI at a ratio of 1:3 (DNA:PEI) in Opti-MEM I (1×) + GlutaMAX. Forty-eight hours post-transfection, the supernatant was transferred onto new plates transfected 24 hr prior with VSV-G. After a further 48 hr, these plates were re-transfected with VSV-G. After 24 hr the resulting PPs were collected, cleared by centrifugation at 2000 × g for 5 min, and stored at −80°C. Subsequent VSV-G PP batches were produced by infecting VSV-G transfected HEK-293T cells with VSV-G PPs at an MOI of 0.1. Titers were determined by preparing 10-fold serial dilutions in Opti-MEM I (1×) + GlutaMAX. Aliquots of each dilution were added to monolayers of $2 \times 10^4$ Vero cells in the same medium in a 96-well plate. Three replicates were performed per PP stock. Plates were incubated at 37°C overnight and then scanned using an Amersham Typhoon scanner. Individual infected cells were quantified using ImageQuant TL software. All PP work was performed in a Class II Biosafety Cabinet under BSL-2 conditions at Erasmus Medical Center.

### Coronavirus S pseudotyped particle production

For the production of SARS-CoV and SARS-CoV-2 S PPs, as well as MBCS mutant PPs, HEK-293T cells were transfected with 15 µg S expression plasmids. Twenty-four hours post-transfection, the medium was replaced for Opti-MEM I (1×) + GlutaMAX, and cells were infected at an MOI of 1 with VSV-G PPs. Two hours postinfection, cells were washed three times with Opti-MEM I (1×) + GlutaMAX and replaced with medium containing anti-VSV-G neutralizing antibody (clone 8G5F11; Absolute Antibody) at a dilution of 1:50,000 to block remaining VSV-G PPs. The supernatant was collected after 24 hr, cleared by centrifugation at 2000 × g for 5 min, and stored at 4°C until use within 7 days. Coronavirus S PPs were titrated on VeroE6 cells as described above.

### Entry route assay

VeroE6, Calu-3, and VeroE6-TMPRSS2 cells were seeded in 24-well plates and kept at 37°C 5% $CO_2$ overnight to achieve 80–100% confluency by the next day. Cells were pretreated with a concentration range of camostat, E64D, or DMSO (with all conditions containing equal concentrations of DMSO) in Opti-MEM I (1×) + GlutaMAX for 2 hr before infecting with on average 1000 PPs per well. Plates were incubated overnight at 37°C 5% $CO_2$ before scanning for GFP signal as described above.

### Entry speed assay

Calu-3 cells were seeded as for entry route assays and pretreated with 10 µM E64D. After 1 hr, PPs were added per well to achieve 1000 infected cells in the control well. At the same time as addition of PPs, 10 µM camostat was added into the first set of wells (t = 0). DMSO was added to controls.

The same inhibitor was added in the next sets of wells in triplicate 2, 4, and 6 hr postinfection. Plates were incubated overnight at 37°C 5% $CO_2$ before scanning for GFP signal as described above.

Authentic virus entry speed was performed in the same manner, by infecting Calu-3 cells with $1 \times 10^4$ TCID50 SARS-CoV-2 and $5 \times 10^4$ TCID50 SARS-CoV. After 12 hr, plates were fixed and blocked as above for transwell inserts. Cells were incubated with mouse-anti-double stranded RNA (Clone J2, 1:500, Scicons) in blocking buffer for 2 hr at room temperature or overnight at 4°C. Cells were washed twice with PBS and stained with Alexa488 conjugated secondary antibody (1:500, Invitrogen) in blocking buffer for an hour at room temperature. Finally, cells were washed twice with PBS and scanned in PBS on the Amersham Typhoon as described above.

## Coronavirus S pseudotyped particle concentration

PPs were concentrated on a 10% sucrose cushion (10% sucrose, 15 mM Tris–HCl, 100 mM NaCl, 0.5 mM EDTA) at 20,000 × g for 1.5 hr at 4°C. Supernatant was decanted, and pellet was resuspended overnight at 4°C in Opti-MEM I (1×) + GlutaMAX to achieve 100-fold concentration. PPs were titrated and aliquots were lysed in 1× Laemmli buffer (Bio-Rad) containing 5% 2-mercaptoethanol for western blot analysis.

## PP infection of primary airway cells

To determine the effect of MBCS mutations on SARS-CoV-2 entry, we obtained airway culture fragments from 2D differentiated bronchiolar cultures as described above. Next, fragments were infected with 40 µL of concentrated wild-type and MBCS mutant PPs for 2 hr at 37°C 5% $CO_2$. Subsequently, the supernatant was replaced with 30 µL BME and plated in a 48-well plate. Approximately 200,000 cells were plated per well. After solidification of the BME, 200 µL PC was added per well and plates were incubated at 37°C 5% $CO_2$. After overnight incubation, the amount of infected cells and organoids per five fields were counted and images taken using a Carl ZEISS Vert.A1 microscope paired with an AxioCam ICm1 camera and Colibri seven laser (469/38 nm for GFP) using ZEN analysis software.

## Spike protein western blot

Concentrated PPs diluted in 4× Laemmli loading buffer were boiled for 30 min at 95°C. S transfected HEK-293T cells were lysed using IP Lysis Buffer (Pierce). Cell lysate was rotated for 30 min and centrifuged for 10 min at 15,000 × g. Supernatant was used for subsequent protein expression analysis. Lysates were diluted in 4× Laemmli loading buffer containing 20% 2-mercaptoethanol and boiled for 30 min at 95°C. PPs and cell lysates were used for sodium dodecyl sulfate (SDS)–polyacrylamide gel electrophoresis analysis using precast 10% TGX gels (Bio-Rad). Gels were run in Tris–glycine–SDS buffer at 50 V for 30 min and subsequently at 120 V for 90 min. Transfer was performed at 300 mA for 55 min onto 0.45µm Immobilon-FL PVDF membranes in Tris–glycine buffer containing 20% methanol. Spike was stained using polyclonal rabbit-anti-SARS-CoV S1 (1:1000, Sino Biological) followed by infrared-labeled secondary antibodies (1:20,000, Licor). All cell lysate western blots are stained for GAPDH using a monoclonal mouse-anti-GAPDH antibody (sc-32233, 1:1000, Santa Cruz Biotechnology) followed by infrared-labeled secondary antibodies. Western blots were scanned on an Odyssey CLx and analyzed using Image Studio Lite Ver 5.2 software.

## Silver staining

All PP western blots had corresponding silver stains performed to assess the quality of the PP preps and for the detection of VSV-N. Samples boiled in Laemmli buffer for western blot analysis were also ran on a 10% w/v gel at 50 V for 30 min followed by 120 V for 90 min before transferring gel into ultrapure water. Silver stains were performed per manufacturer's instructions using the Silver Stain for Mass Spectrometry Kit (Pierce). Colorimetric images were taken on the Amersham AI600 (GE Healthcare).

## Statistical analysis

Statistical analysis was performed with the GraphPad Prism 5 and 8 software using a t-test, one-way ANOVA, or two-way ANOVA followed by a Bonferroni multiple-comparison test.

## Acknowledgements

This work was supported by NWO Grant 022.005.032, partly financed by the Netherlands Organization for Health Research and Development (ZONMW) grant agreement 10150062010008 to BLH and co-funded by the PPP Allowance (grant agreement LSHM19136) made available by Health Holland, Top Sector Life Sciences and Health, to stimulate public–private partnerships. The present manuscript was part of the research program of the Netherlands Centre for One Health.

## Additional information

### Funding

| Funder | Grant reference number | Author |
|---|---|---|
| Nederlandse Organisatie voor Wetenschappelijk Onderzoek | 0.22.005.032 | Bart L Haagmans |
| ZonMw | 10150062010008 | Bart L Haagmans |
| Health Holland | LSHM19136 | Bart L Haagmans |

The funders had no role in study design, data collection and interpretation, or the decision to submit the work for publication.

### Author contributions

Anna Z Mykytyn, Tim I Breugem, Conceptualization, Formal analysis, Validation, Investigation, Visualization, Methodology, Writing - original draft, Writing - review and editing; Samra Riesebosch, Formal analysis, Investigation, Visualization, Writing - review and editing; Debby Schipper, Petra B van den Doel, Investigation, Writing - review and editing; Robbert J Rottier, Resources, Writing - review and editing; Mart M Lamers, Conceptualization, Formal analysis, Supervision, Validation, Investigation, Visualization, Methodology, Writing - original draft, Writing - review and editing; Bart L Haagmans, Conceptualization, Supervision, Funding acquisition, Writing - review and editing

### Author ORCIDs

Anna Z Mykytyn (iD) https://orcid.org/0000-0001-7188-6871
Tim I Breugem (iD) https://orcid.org/0000-0002-5558-7043
Robbert J Rottier (iD) http://orcid.org/0000-0002-9291-4971
Mart M Lamers (iD) https://orcid.org/0000-0002-1431-4022
Bart L Haagmans (iD) https://orcid.org/0000-0001-6221-2015

### Decision letter and Author response

Decision letter https://doi.org/10.7554/eLife.64508.sa1
Author response https://doi.org/10.7554/eLife.64508.sa2

## Additional files

### Supplementary files

• Transparent reporting form

### Data availability

All data generated or analysed during this study are included in the manuscript and supporting files.

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
