## [Decision Letter]

**Acceptance summary:**

SARS-CoV-2 targets the human airway epithelium to establish infections that frequently lead to COVID-19. This paper describes a three-dimensional human airway organoid (huAO) system to study SARS-CoV-2 cellular infection. The authors demonstrate that efficient entry involves cleavage of the viral spike (S) protein by host serine proteinases and that the multibasic cleavage site (MBCS) in the SARS-CoV-2 S protein increases the efficiency of entry of this virus compared to viruses lacking a MBCS. The study provides important insights into the mechanism of virus entry in a relevant cell system.

**Decision letter after peer review:**

Thank you for submitting your article "SARS-CoV-2 entry into human airway organoids is serine protease-mediated and facilitated by the multibasic cleavage site" for consideration by *eLife*. Your article has been reviewed by three peer reviewers, one of whom is a member of our Board of Reviewing Editors, and the evaluation has been overseen by Miles Davenport as the Senior Editor. The reviewers have opted to remain anonymous.

The reviewers have discussed the reviews with one another and the Reviewing Editor has drafted this decision to help you prepare a revised submission.

Coronavirus entry into cells requires membrane fusion that is triggered by proteolytic cleavage of the viral spike (S) protein. In addition to the consensus S2' cleavage site, SARS-CoV-2 contains an additional cleavage site, designated S1/S2, that is not found in SARS-CoV-1. The paper presents data that address the impact of this second site on virus entry. The work extends findings from other groups and, importantly, uses primary respiratory epithelial cells as a more relevant model SARS-CoV-2 infection.

Overall, the reviewers consider that the paper makes an important contribution to knowledge of SARS-CoV-2 entry into cells, however they do raise a number of points that you should address to clarify aspects of the work. In particular, with regard to the primary cell system (reviewers 1 and 2), can you explain why it was necessary to disrupt the differentiated cell systems prior to infection and whether the apparent viral tropism for ciliated cells is evident in the cells seen to be infected in the organoids?

Reviewers 1 and 3 both raise the question as to whether the altered entry kinetics seen for viruses containing the MBCS might be explained by interaction of S1/S2 cleaved S with neuropilins. Some discussion of this possibility is certainly warranted.

Reviewer #1:

This study aims to test the relative importance of cell surface serine proteases (TMPRSS2) versus endo/lysosomal cathepsins in the entry of SARS coronaviruses (SARS-CoV and SARS-CoV-2) into primary respiratory epithelial cells. The fusion activity of coronaviruses is activated by proteolytic cleavage of the viral S protein at a site designated S2'. Interestingly, the S protein of SARS-CoV-2 contains a small insertion that creates a multibasic cleavage site (MBCS) i.e., a second cleavage site (designated S1/S2) that is a target for cellular furin-like proteases. Published experiments in CaLu (human lung cancer derived) and Vero (African green monkey kidney) cells have suggested the MBCS increases the efficiency of SARS-CoV-2 fusion and entry, though a mechanism has not been established. The work described in this paper attempts to determine whether S1/S2 cleavage influences the ability of SARS-CoV-2 to infect more relevant cell cultures established from primary human respiratory tissue.

To this end, the authors generate SARS-CoV S protein mutants in which the MBCS is inactivated by removal of four amino acids, or mutation of key residues in the MBCS. The ability of pseudotyped viruses carrying the various S proteins, as well as SARS-CoV and SARS-CoV-2, is then tested in CaLu, Vero and primary human airway cultures. The data support published work indicating that the SARS-CoV-2 MBCS facilitates efficient cellular entry and, in relevant cell systems i.e., CaLu and primary cell cultures, efficient infection appears to rely on S protein activation by cell surface serine proteases. The study emphasises the importance of using appropriate tissue models for understanding the details of viral replication. However, I have some reservations about the work and its presentation that warrant attention.

The authors show some characterisation of their primary cell culture system in Figure 1—figure supplement 1. Here the cells are grown at an air liquid interface and appear well differentiated with multiple cell types present. They examine the distribution of two key cellular factors for SARS-CoV-2 entry using antibody labelling. Firstly, they have used HRP detection to locate the proteins. In my view this is less sensitive than immunofluorescence and I wonder why they have used the HRP technique. Secondly, the labelling indicates the two proteins have different locations: ACE2 at the surface of the tissue and TMPRSS2 deeper within the tissue. The authors do not comment on this or its implications for virus entry. For infection, these differentiated tissues are disrupted, the separated cells incubated with virus and the cells then allowed to form organoids (HAO). These organoids are not well characterised; it is not at all clear what cell types are infected, or why it is necessary to use this approach rather than test infection on the differentiated tissue (as in Figure 1—figure supplement 1), which appears to have an architecture similar to that of normal respiratory epithelial tissue.

The authors frequently state that the MBCS increases the rate of fusion or fusogenicity. Although fusion is key for virus entry, the event that is indirectly analysed is S protein S2' cleavage. It would be more accurate to indicate this than “rate of fusion”. The authors discuss the notion that the MBCS alters the accessibility of the S2' cleavage site, which may indeed be the case. Another contributing factor may be the ability of the S1/S2 cleaved SARS-CoV-2 S1 to bind neuropilin 1/2 through a C-end rule motif generated by cleavage of the MBCS, as suggested by two recent papers in Science. The authors should at least comment on this and whether the cell lines and primary cells used in this study express neuropilins 1/2.

In Figures 4A/B the authors show that both SARS-CoV and SARS-CoV-2 PPS infect Vero cells by the cathepsin pathway and entry is not reduced by Camostat. By contrast, in Vero cells expressing TMPRSS2 (Figure 4C/D), infection by SARS-CoV-2 PPs is significantly inhibited by Camostat and not by E64D. Why is this? The endosomal pathway should still be active in these cells.

Reviewer #2:

This manuscript describes a series of experiments in cell lines and primary airway cells showing that the presence of a multibasic cleavage site (MBCS) in SARS-CoV2 spike protein, which is absent in the SARS-CoV spike, was responsible for increased infectivity in human airway organoids and human cell line (Calu3), but not VeroE6 cells. The authors demonstrate that infectivity of these cells could be inhibited by serine protease inhibitors (camostat).

The manuscript is well written and reads easily. The premise of the study – that the MBCS site is key to the differences in SARS-CoV and SARS-CoV2 infectivity is novel and would be an addition to the field, warranting publication. This reviewer particularly liked how the authors speculated as to this maybe why chloroquine, which acts in the endosome, maybe ineffective at blocking infection in serine protease expressing cells. As lung cells express serine proteases, inhibitors that block endosomal entry are likely to be ineffective in vivo. This builds nicely on the argument that we should be using more relevant human models if we want to develop effective therapeutics.

Reviewer #3:

The authors investigate if the SARS-CoV-2 Spike MBCS (multibasic cleavage site) affects viral entry and its dependence on host proteases TMPRSS2 and Cathepsin L. SARS-CoV-1 Spike does not have a MBCS. They show that the MBCS promotes virus infection into Calu3 cells whereas viruses lacking it prefers VeroE6 cells. Using transduction, infection, physiological airway cell cultures and quantitative syncytia assays etc. they show that the MBCS leads to faster cell entry in a TMPRSS2-dependent fashion. This can be inhibited by the clinically approved inhibitor camostat mesylate. It is a focused and informative study that extends the observations of Hoffmann et al., 2020a and others.

1) The authors emphasise that the MBCS is a viral tactic for improved entry into airway human cells. Other human cells were not tested, so I think this is a leap in logic, and others have clearly shown that MBCS-dependence is not limited to human airway cells.

Examples are; wt SARS-CoV-2 S compared to the MBCS-deficient S infected intestinal HEK and Caco-2 cells better than VeroE6. Infectivity of wt virus increased under expression of TMPRSS2 and neuropilin-1 (Cantuti-Castelvetri et al., 2020, Figure 1E and Figure 3—figure supplement 1). Furthermore, neuropilin-1 binding the S1 C-end rule motif (liberated by MBCS cleavage of SARS-CoV-2 S) enhances infectivity and spread of authentic virus in Caco-2 cells (Daly et al., 2020). These points should be discussed in context of the authors' findings.

2) Along the same lines as 1), could the authors test Caco-2 cells?

3) What are the protein (or mRNA) expression levels of TMPRSS2 and CathepsinL in the Vero, Vero-TMPRSS2, Calu3 (and Caco2) cells?

4) Figure 3 and Figure 3 —figure supplement 1B-D. It would be useful to see quantification of the enhanced syncytia in VeroE6-TMPRSS2 cells compared to VeroE6.

---

## [Author Response]

Coronavirus entry into cells requires membrane fusion that is triggered by proteolytic cleavage of the viral spike (S) protein. In addition to the consensus S2' cleavage site, SARS-CoV-2 contains an additional cleavage site, designated S1/S2, that is not found in SARS-CoV-1. The paper presents data that address the impact of this second site on virus entry. The work extends findings from other groups and, importantly, uses primary respiratory epithelial cells as a more relevant model SARS-CoV-2 infection.Overall, the reviewers consider that the paper makes an important contribution to knowledge of SARS-CoV-2 entry into cells, however they do raise a number of points that you should address to clarify aspects of the work. In particular, with regard to the primary cell system (reviewers 1 and 2), can you explain why it was necessary to disrupt the differentiated cell systems prior to infection and whether the apparent viral tropism for ciliated cells is evident in the cells seen to be infected in the organoids?

We acknowledge that the requirement for this step may not be explained well enough in the manuscript and we have clarified the rationale behind this in the text.

“A 3D airway organoid model that allows efficient SARS-CoV-2 replication where infectious virus titers increase over time has not yet been established, but we previously reported that 2D airway organoid-derived air-liquid interface differentiated cultures allow efficient SARS-CoV-2 replication (Lamers et al., 2020). […] To setup a 3D model, we dissociated these 2D air-liquid differentiated cultures into small clumps, infected these in suspension and then re-plated the clumps into basement membrane extract (BME), in which they formed spheroids. SARS-CoV-2 PPs successfully infected these hAOs, as observed by fluorescent microscopy (Figure 1E).”

The tropism for ciliated cells is also observed in the 3D spheroids. This was investigated in Figure 5B, where we performed co-stains of AcTUB and viral nucleoprotein. In these pictures the majority of infected cells appear ciliated. However, ciliated cells are not the only cells infected by SARS-CoV-2. Non-ciliated cells can also be a target, as we observed club cell infection in 2D ALI cultures (Lamers et al., 2020 EMBO J).

Reviewers 1 and 3 both raise the question as to whether the altered entry kinetics seen for viruses containing the MBCS might be explained by interaction of S1/S2 cleaved S with neuropilins. Some discussion of this possibility is certainly warranted.

Recent reports indeed indicate that neuropilins (NRP) may be an additional entry factor for SARS-CoV-2 (Cantuti-Castelvetri et al., 2020; Daly et al., 2020). This is an interesting hypothesis as ACE2 expression (mRNA and protein) is relatively low while NRP expression appears to be abundant in almost all pulmonary cells.

We have discussed this in the text:

“Recent reports indicate that neuropilins (NRP) may be an additional entry factor for SARS-CoV-2 (Cantuti-Castelvetri et al., 2020; Daly et al., 2020). […] Structural changes caused by S1/S2 cleavage may affect protease accessibility or protein interactions at the cell membrane as well as increase subsequent S2’ cleavage.”

Reviewer #1:This study aims to test the relative importance of cell surface serine proteases (TMPRSS2) versus endo/lysosomal cathepsins in the entry of SARS coronaviruses (SARS-CoV and SARS-CoV-2) into primary respiratory epithelial cells. The fusion activity of coronaviruses is activated by proteolytic cleavage of the viral S protein at a site designated S2'. Interestingly, the S protein of SARS-CoV-2 contains a small insertion that creates a multibasic cleavage site (MBCS) i.e., a second cleavage site (designated S1/S2) that is a target for cellular furin-like proteases. Published experiments in CaLu (human lung cancer derived) and Vero (African green monkey kidney) cells have suggested the MBCS increases the efficiency of SARS-CoV-2 fusion and entry, though a mechanism has not been established. The work described in this paper attempts to determine whether S1/S2 cleavage influences the ability of SARS-CoV-2 to infect more relevant cell cultures established from primary human respiratory tissue.To this end, the authors generate SARS-CoV S protein mutants in which the MBCS is inactivated by removal of four amino acids, or mutation of key residues in the MBCS. The ability of pseudotyped viruses carrying the various S proteins, as well as SARS-CoV and SARS-CoV-2, is then tested in CaLu, Vero and primary human airway cultures. The data support published work indicating that the SARS-CoV-2 MBCS facilitates efficient cellular entry and, in relevant cell systems i.e., CaLu and primary cell cultures, efficient infection appears to rely on S protein activation by cell surface serine proteases. The study emphasises the importance of using appropriate tissue models for understanding the details of viral replication. However, I have some reservations about the work and its presentation that warrant attention.The authors show some characterisation of their primary cell culture system in Figure 1—figure supplement 1. Here the cells are grown at an air liquid interface and appear well differentiated with multiple cell types present. They examine the distribution of two key cellular factors for SARS-CoV-2 entry using antibody labelling. Firstly, they have used HRP detection to locate the proteins. In my view this is less sensitive than immunofluorescence and I wonder why they have used the HRP technique.

We have used the HRP technique for the detection of ACE2 and TMPRSS2 because these stainings had already been optimized using this technique previously. The advantage of this technique is that the cells can be counterstained using hematoxylin to see the background of the cell (such as cilia or mucus-filled vesicles of goblet cells). We have not directly compared the sensitivity of our ACE2 and TMPRSS2 stainings with immunofluorescence and HRP.

Secondly, the labelling indicates the two proteins have different locations: ACE2 at the surface of the tissue and TMPRSS2 deeper within the tissue. The authors do not comment on this or its implications for virus entry.

We acknowledge that this is an interesting point, but we have not commented on this to avoid speculation. Although our stainings indicate that TMPRSS2 may be more localized to cytoplasmic secretory vesicles, this does not exclude that there is TMPRSS2 as well on the apical surface (as has been shown by others). In addition, early TMPRSS2-mediated entry may not be limited to the apical plasma membrane, but could also occur in vesicles that have just been endocytosed, but have not yet become acidified endosomes. This would also be regarded as “early” entry.

For infection, these differentiated tissues are disrupted, the separated cells incubated with virus and the cells then allowed to form organoids (HAO). These organoids are not well characterised; it is not at all clear what cell types are infected, or why it is necessary to use this approach rather than test infection on the differentiated tissue (as in Figure 1—figure supplement 1), which appears to have an architecture similar to that of normal respiratory epithelial tissue.

Indeed this 3D organoid infection model in which cells are differentiated at ALI is novel. However, the cultures are not separated completely into single cells. Instead, they are disrupted into small clumps which form organoids within minutes at 37 degrees celsius in culture medium. As explained above, we have chosen this approach as there is currently no established 3D differentiation protocol for airway organoids to study SARS-CoV-2 replication. For certain purposes 3D airway models may be preferred over 2D models, which is why we used both in Figure 5. In addition, a single ALI 2D organoid culture well (12 well format) can be used to plate approximately 10 3D organoid wells, as the cells grow at a high density at ALI, allowing us to perform more experiments with limited material.

In the 2D ALI cultures, we mostly see infection of ciliated cells, but club cells are occasionally infected as well (Lamers et al., 2020 Science; Lamers et al., 2020 EMBO J). Although changes in the tropism of SARS-CoV-2 are not expected when these cultures are plated in 3D, we did perform co-stainings of AcTub and viral nucleoprotein to show that ciliated cells are infected in this model as well (Figure 5B).

The authors frequently state that the MBCS increases the rate of fusion or fusogenicity. Although fusion is key for virus entry, the event that is indirectly analysed is S protein S2' cleavage. It would be more accurate to indicate this than “rate of fusion”. The authors discuss the notion that the MBCS alters the accessibility of the S2' cleavage site, which may indeed be the case. Another contributing factor may be the ability of the S1/S2 cleaved SARS-CoV-2 S1 to bind neuropilin 1/2 through a C-end rule motif generated by cleavage of the MBCS, as suggested by two recent papers in Science. The authors should at least comment on this and whether the cell lines and primary cells used in this study express neuropilins 1/2.

We use the term fusion, and not S2’ cleavage because we see S2’ cleavage as a biochemical term that should be measured in a biochemical assay. In addition, under certain conditions fusion may occur without S2’ cleavage at auxiliary sites that have yet to be identified. Therefore, we are more comfortable with the term fusion.

We agree that the role of NRPs warrants discussion and have added this in Discussion paragraph three. We have unpublished mRNA sequencing data that NRP1 is expressed in the organoids used in this study.

In Figures 4A/B the authors show that both SARS-CoV and SARS-CoV-2 PPS infect Vero cells by the cathepsin pathway and entry is not reduced by Camostat. By contrast, in Vero cells expressing TMPRSS2 (Figure 4C/D), infection by SARS-CoV-2 PPs is significantly inhibited by Camostat and not by E64D. Why is this? The endosomal pathway should still be active in these cells.

Indeed the endosomal entry pathway is still active in these cells, but the infectivity of PPs is increased approximately 10 fold by ectopic TMPRSS2 expression. Therefore, the relative entry left after camostat treatment is around 10% (Figure 4). Likewise, the decrease in entry by E64D is expected to be around 10%.

Reviewer #3:The authors investigate if the SARS-CoV-2 Spike MBCS (multibasic cleavage site) affects viral entry and its dependence on host proteases TMPRSS2 and Cathepsin L. SARS-CoV-1 Spike does not have a MBCS. They show that the MBCS promotes virus infection into Calu3 cells whereas viruses lacking it prefers VeroE6 cells. Using transduction, infection, physiological airway cell cultures and quantitative syncytia assays etc. they show that the MBCS leads to faster cell entry in a TMPRSS2-dependent fashion. This can be inhibited by the clinically approved inhibitor camostat mesylate. It is a focused and informative study that extends the observations of Hoffmann et al., 2020a, and others.1) The authors emphasise that the MBCS is a viral tactic for improved entry into airway human cells. Other human cells were not tested, so I think this is a leap in logic, and others have clearly shown that MBCS-dependence is not limited to human airway cells.Examples are; wt SARS-CoV-2 S compared to the MBCS-deficient S infected intestinal HEK and Caco-2 cells better than VeroE6. Infectivity of wt virus increased under expression of TMPRSS2 and neuropilin-1 (Cantuti-Castelvetri et al., 2020, Figure 1E and Figure 3—figure supplement 1). Furthermore, neuropilin-1 binding the S1 C-end rule motif (liberated by MBCS cleavage of SARS-CoV-2 S) enhances infectivity and spread of authentic virus in Caco-2 cells (Daly et al., 2020). These points should be discussed in context of the authors' findings.

Indeed we have only looked at human airway cells and therefore our conclusions are only on airway cells. However, it is indeed possible that the MBCS increases entry into other cells as well, and we briefly discussed this in the text.

A discussion on neuropilins has been added as well in Discussion paragraph three.

2) Along the same lines as 1), could the authors test Caco-2 cells?

We believe that it is beyond the current study to include Caco-2 cells, which is a transformed cell line of the intestinal epithelium. The relevance of these cells in understanding the pathogenesis of COVID-19 is debatable.

3) What are the protein (or mRNA) expression levels of TMPRSS2 and CathepsinL in the Vero, Vero-TMPRSS2, Calu3 (and Caco2) cells?

The expression levels of these cells have been extensively quantified by others. Calu-3 cells have a low Cathepsin L expression but high TMPRSS2 expression. VeroE6 cells express cathepsins to high levels but do not express TMPRSS2 (Shirato, 2013; Bertram et al., 2010; Limburg et al., 2019; Kawase et al., 2012).

4) Figure 3 and Figure 3—figure supplement 1B-D. It would be useful to see quantification of the enhanced syncytia in VeroE6-TMPRSS2 cells compared to VeroE6.

Done. We added both the quantification of the fusogenicity of SARS-CoV-2 and SARS-CoV S on VeroE6 and VeroE6-TMPRSS2 (Figure 3—figure supplement 1E and 1F). Additionally, fold change in total fusion due to TMPRSS2 overexpression was quantified in Figure 3—figure supplement 1G. For both SARS-CoV and SARS-CoV-2 S the expression of TMPRSS2 increase fusion ~2 fold (subsection “Cell-cell fusion is facilitated by the SARS-CoV-2 multibasic cleavage site and SARS-CoV-2 is more fusogenic than SARS-CoV on human airway organoids”).